# Well-being as a function of person-country fit in human values

Paul H. P. Hanel [1,2 ✉], Uwe Wolfradt[3], Lukas J. Wolf [1,4], Gabriel Lins de Holanda Coelho[4,5] & Gregory R. Maio[1]

It is often assumed that incongruence between individuals' values and those of their country is distressing, but the evidence has been mixed. Across 29 countries, the present research investigated whether well-being is higher if people's values match with those of people living in the same country or region. Using representative samples, we find that person-country and person-region value congruence predict six well-being measures (e.g., emotional well-being, relationship support; N = 54,673). Crucially, however, value type moderates whether person-country fit is positively or negatively associated with well-being. People who value self-direction, stimulation, and hedonism more and live in countries and regions where people on average share these values report lower well-being. In contrast, people who value achievement, power, and security more and live in countries and regions where people on average share these values, report higher well-being. Additionally, we find that people who moderately value stimulation report the highest well-being.

[1] Department of Psychology, University of Bath, Bath BA2 7AY, UK. [2] Department of Psychology, University of Essex, Colchester CO4 3SQ, UK. [3] Institut für Psychologie, Martin-Luther-Universität Halle-Wittenberg, 06099 Halle, (Saale), Germany. [4] School of Psychology, Cardiff University, Cardiff CF10 3AT, UK. [5] School of Applied Psychology, University College Cork, Cork T12 K8AF, Ireland. ✉email: p.hanel@essex.ac.uk

Scholars have long argued that shared values are important for well-being. In theory, shared values bring in a common framework that increases the subjective sense of connection to one's culture and alignment of personal motives with social/environmental norms[1–5]. The present paper critically re-examines this longstanding perspective using international data.

In psychological research, human values are usually defined as abstract ideals that guide actions and express needs[6–8]. One of the most prominent models of human values distinguishes ten value types: self-direction, universalism, benevolence, tradition, conformity, security, power, achievement, hedonism, and stimulation[8]. Schwartz's model predicts a circumplex pattern of correlations between ratings of value importance (Fig. 1), based on their motivational conflicts and compatibilities, and this pattern has been confirmed in over 80 countries across three decades of research[9]. The model has also been supported using a range of experimental analyses of judgments and behavior, and brain structure[10,11].

The values described in this model can be examined at the level of individuals and aggregate groups, such as countries, and congruence between personal and national values has theoretical implications for personal well-being. Four potential reasons for this link are environmental affordances, social sanctions, internal conflict, and shared realities[12–17]. For example, if an individual's society values achievement (e.g., success), and the individual does as well, then the individual might feel supported by the sense of common purpose. If the individual does not value achievement, the individual might feel pressured by social norms to achieve more. This pressure, in turn, may reduce the individual's subjective well-being. Further, incongruence between people's own and their compatriots' values can result in internal conflicts, which may result in reduced well-being. This reduction might occur because people's goals are blocked in incongruent environments. Conversely, goal facilitation may occur in congruent value environments. Moreover, congruence in values can result in the experience of shared reality and identity, which in turn helps to validate one's experience, facilitates cooperation, coordination, and promotes a sense of belonging[14,18].

Yet, past research has been inconclusive about whether value congruence (sometimes framed as discrepancy) predicts well-being[12,15,17,19–25]. This inconclusiveness may arise because of several methodological and analytical differences across studies, which makes it difficult to compare the studies. For example, studies have conceptualized person–environment value congruence with either an objective or a subjective approach[12,22]. In the objective approach, individuals' self-rated value priorities are compared with the actual value priorities of others (e.g., family, fellow students, and society). In the subjective approach, the self-rated priorities are compared with the individual's perceptions of the value priorities of others. In addition, there are differences in the type of sample used: students or nonstudent samples. Further, the statistical approaches differ, which may also have a strong impact on the outcome. Some researchers used difference scores (e.g., between own and perceived values), while others used profile correlations as an estimate of congruency. To obtain profile correlations, the importance a person places on all value types (e.g., all ten value types of Schwartz's model; cf. Fig. 1) is correlated with the country average. This correlation coefficient that is obtained for each participant is used as an estimate of value congruence and correlated with their well-being score.

However, both difference scores and profile correlations have been challenged for multiple reasons. For example, difference scores have been criticized as "often less reliable than either of their component measures… [and] inherently ambiguous, given that they combine measures of conceptually distinct constructs into a single score"[26]. In other words, difference scores reduce a three-dimensional relation between the predictors (e.g., own and other people's values) and the outcome (e.g., well-being) to a ambiguous two-dimensional relation. Profile correlations have been criticized as "conceptually ambiguous, discard information essential to testing congruence hypotheses, conceal the source of the difference between entities…"[27]. That is, profile correlations mask differences in similarity effects between value types because they must be computed across several value types. These issues greatly limit the strength of conclusions from previous research of person–environment value congruence.

As a superior alternative to difference scores and profile correlations, Edwards[26,27] suggested using polynomial regressions and response surface analysis. These methods model the relations between two predictors (e.g., own values and the values of one's own country) and an outcome (e.g., well-being) in a three-dimensional space. This approach enables them to provide more information than alternative approaches, such as difference scores and moderated regression[28]. For example, the complex interplay between the two predictors can be visually displayed using response surface analysis. This plot displays a surface with the estimated outcome of all possible combinations of the predictors. Such plots allow researchers to easily identify the location of the similarity effect (e.g., is well-being higher when both a person and other people same in the country they are living in value security above average?). This utility is illustrated by Bleidorn et al.'s[29] use of these methods to investigate person–environment fit in personality traits. Across 860 US-American cities, they found that people had higher self-esteem if their personality traits aligned with the average traits of the city in which they lived. However, the effect sizes were very small and only reliable for some personality traits (e.g., openness and agreeableness). Similarly, polynomial regressions have been used to examine effects of person–environment fit in traits on indicators of entrepreneurial success[30], attachment[31], as well as fit in values on anti-immigrant attitudes[32] and national pride[33]. These studies have demonstrated the utility of the polynomial method as a powerful tool for precisely estimating effects of person–environment fit.

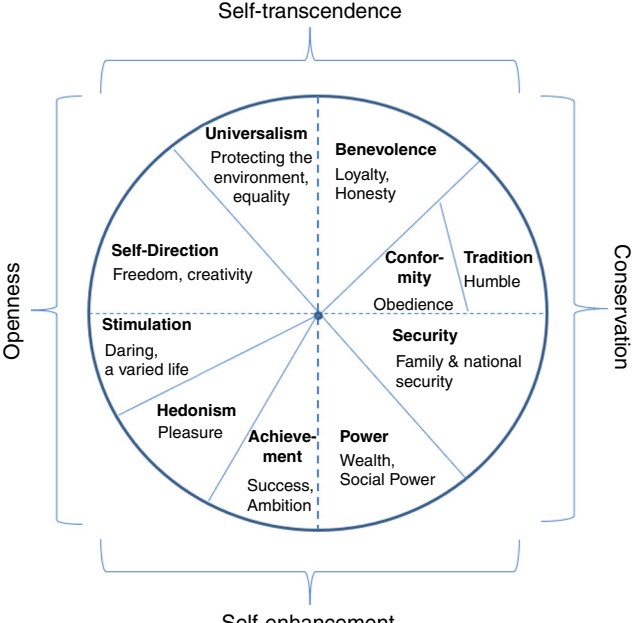

**Fig. 1 Model of human values.** Schwartz'[8] quasi-circumplex model of human values displaying four higher value types, ten value types (bold font), and examples of values in each type (normal font; adapted from ref. [75]).

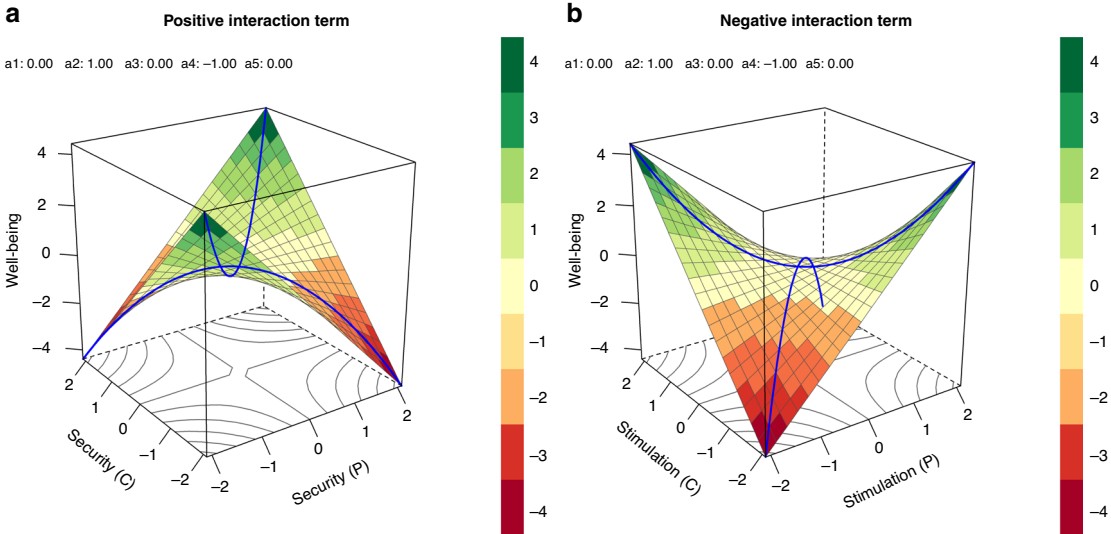

**Fig. 2 Predicted associations between values and well-being.** Predicted pattern of results illustrated for security (positive interaction term (**a**)) and stimulation (negative interaction term (**b**)), with personal (P) and country (C) level values as predictors of well-being. Both graphs display interaction terms alone in the absence of linear and quadratic effects. Source data are provided as a Source data file.

We chose to use the polynomial methodology to disambiguate the connection between person–country value congruence and subjective well-being. This method enables an examination of the role of value congruence independently of the linear effects of personal values on well-being[34,35]. This method takes into account the linear and quadratic effects of a person's own values, the linear and quadratic effects of the environment (e.g., country or region), and the interaction between a person's values and an environment's values (i.e., value congruence). The quadratic terms, the two linear terms, and the interaction are necessary to test for similarity effects robustly[36]. Polynomial regressions allow us to investigate these effects together, which will enable a more conclusive overview of the relations between values and well-being.

To make our application of polynomial methods relevant across diverse regions and countries, we used data from 28 European countries and Israel. We further expanded previous research by including six variables related to well-being[37]: evaluative well-being (overall satisfaction with life and happiness), emotional well-being (positive feelings and lack of negative feelings in the past week), functioning (e.g., self-esteem, optimism, and feelings of autonomy), vitality (e.g., feeling energized and quality of sleep), community well-being (feeling close to and trusting other people, especially in the own community), and supportive relationships (feeling appreciated and supported by other people). While most previous studies focused on only one dependent variable, including several measures of well-being allows a better understanding of the processes of person–environment fit. For instance, does fit in values relate more strongly to personal well-being outcomes (e.g., evaluative and emotional well-being) or interpersonal well-being outcomes (e.g., community well-being and supportive relationships)? This question is difficult to address using prior studies because differences in predictors and statistical analyses prevent straightforward comparisons of effect sizes. Also, having several well-being variables allows us to replicate and extend previous research. For example, we can test whether value fit is associated with self-esteem in the same manner as person–region trait fit correlates with self-esteem[29].

Of particular theoretical importance, the use of Schwartz's[8] model allows a fine-grained analysis for which types of values yield effects of person–environment fit on well-being. As outlined

above, it is plausible that person–country fit in all values predicts well-being, but a crucial issue is whether these relations depend on value type. Past research examining value congruence and measures of well-being (e.g., Beilmann and Lilleoja[19] have assessed congruence across values using profile correlations). Because this method aggregates across all values, it does not reveal for which value type value congruence matters most, and whether the effects of congruence vary between different values. A polynomial regression approach allows us to examine each value type separately and to avoid the aforementioned statistical limitations of reliance on profile correlations.

It is conceivable that well-being is supported by person–country value congruence, regardless of the type of value being examined, because the environmental affordances and social sanctions operate similarly for all values. For example, people whose values deviate from the prevalent values in a society might be socially sanctioned, regardless of value type. Alternatively, Schwartz's[8,38] model predicts that values' associations with other variables follow a sinusoidal pattern across values, such that values that serve opposing motivations in his circumplex model will exhibit an opposite pattern of correlations, with other variables. From this perspective, value congruence at opposing ends of the value circle may exhibit opposing patterns of relations with well-being.

As a starting point, we predict that people with high conservation values (e.g., security, tradition, and conformity), which preserve the status quo, would report higher well-being when these values are shared by others in their country because this will make their life more stable. Also, people valuing conservation values identify more with their country[39] and prefer to be similar to people around them[40], which likely results in higher well-being. This prediction is illustrated for security values in Fig. 2, left panel, which shows significant interaction terms in absence of linear/quadratic effects: if the security values of a person ($x$-axis) and their country ($y$-axis) are both high, well-being is highest (see point $x = 2$, $y = 2$, and $z = 4$). In contrast, if people place very low importance on security, they may feel more constrained in a country wherein most people cherish these values (point $-2|2|-4$ in Fig. 2, left panel) and freer if others also do not cherish these values (point $-2|-2|4$ in Fig. 2, left panel). Finally, if a person values security, but others do not, they may feel less safe and unsettled, resulting in lower well-being (point $2|-2|-4$ in Fig. 2,

left panel). Importantly, the linear and quadratic terms qualify these results. For instance, quadratic effects would lift or depress the plane at the 0|0 point, and linear terms would tilt the plane.

However, Schwartz's model raises the possibility that the effects of openness values (hedonism, stimulation, and self-direction) show the opposite pattern of the findings for conservation values. That is, people who consider hedonism, stimulation, and self-direction values to be more important may report higher well-being when others around them consider these values less important. Indeed, people high in openness values identify less with their country[39] and seek to be distinct[40]. Thus, it is conceivable that people who value openness prefer if people around them are less similar to them, which result in higher well-being (point 2|−2|4 in Fig. 2, right panel) than if people around them are similar (point 2|2|−4 in Fig. 2, right panel).

Other research questions focused on the linear simple effects of values. Past research has established reliable positive links between hedonism, stimulation, and self-direction values and well-being[15,34]. These values are also called growth values[38] and promote well-being directly by supporting self-actualization and growth needs[34]. Further, on a country-level, individualism, which entails self-direction and stimulation, is positively associated with well-being[41]. We expect to replicate these findings. We made no predictions regarding quadratic effects. We include a brief discussion of linear and quadratic effects in our "Results" section to obtain a more coherent picture of the relations between values and well-being; the main focus of this paper, however, is on congruence effects.

Finally, we include tests to enable a deeper understanding of the role of social/physical proximity in the value congruence effects. That is, we test whether individual–country or individual–region fit matters more. This issue is relevant because people might identify more with the country (UK or Germany) than the region (e.g., Scotland or Bavaria) in which they live or vice versa.

We made no prediction regarding self-enhancement and self-transcendence values due to the complex and diverging literature related to agentic and communal traits, social comparison, and self-esteem. For example, on the one hand, one might assume that people high in competitiveness would thrive in a competitive environment, while those high in cooperativeness would thrive in a cooperative environment. These environments would be functional for their goals and the self-enhancement and self-transcendence values that activate them. On the other hand, evidence suggests that people socially compare on both agentic and communal traits[42], which may suggest lower well-being when people seem too similar in relatively agentic and communal values, particularly if these are important dimensions of comparison for social identity[43].

We report evidence that value congruence is associated with well-being. Crucially, however, value type moderates whether person–country fit is positively or negatively associated with well-being. People who value self-direction, stimulation, and hedonism more, and live in countries and regions where people on average share these values report lower well-being. In contrast, people who value security more and live in countries and regions where people on average share these values, report higher well-being.

## Results

**Individual–country fit.** We first examined the interactions between individual- and country-level values and, if significant at .001, whether the effects are consistent across well-being type (i.e., (non-)significant for the same value type). Table 1 reports the unstandardized coefficients of the interaction terms. Detailed results that include the coefficients of the linear terms, quadratic

**Table 1 Unstandardized coefficients of the interaction terms.**

| | Evaluative WB | | Emotional WB | | Functioning | | Vitality | | Communal WB | | Support RS | |
|---|---|---|---|---|---|---|---|---|---|---|---|---|
| | C | R | C | R | C | R | C | R | C | R | C | R |
| Security | 0.64*** | 0.37*** | 0.96*** | 0.73*** | 1.47*** | 0.94*** | 1.02*** | 0.78*** | 0.53*** | 0.49*** | 1.72*** | 1.18*** |
| Tradition | 0.10 | 0.23 | −0.04 | 0.19 | 0.40* | 0.54*** | −0.10 | 0.14 | 0.21 | 0.54*** | 0.57*** | 0.70*** |
| Conformity | 0.12 | −0.02 | −0.24 | −0.01 | 0.20 | 0.34* | −0.17 | 0.01 | 0.37* | 0.36* | 0.61*** | 0.46** |
| Benevolence | 0.15 | −0.09 | 0.44* | 0.62*** | 0.07 | −0.16 | 0.49** | 0.60** | −0.43* | −0.26 | −0.14 | −0.15 |
| Universalism | −0.57* | −0.40 | −0.58* | −0.19 | −0.75** | −0.83*** | 0.03 | 0.16 | 0.30 | 0.54* | −1.50*** | −1.40*** |
| Self-direction | −1.85*** | −1.43*** | −1.01*** | −0.31 | −1.85*** | −1.17*** | −0.84*** | −0.51* | −0.40* | −0.14 | −1.44*** | −1.26*** |
| Stimulation | −1.99*** | −1.40*** | −1.29*** | −0.76*** | −1.30*** | −1.06*** | −1.11*** | −0.59*** | −0.17 | −0.07 | −0.63*** | −0.55*** |
| Hedonism | −0.95*** | −0.81*** | −0.14 | −0.03 | −0.32* | −0.36** | −0.27* | −0.34** | −0.06 | 0.06 | −0.08 | −0.22* |
| Achievement | 1.15*** | 0.88*** | 1.56*** | 1.44*** | 1.76*** | 1.52*** | 1.34*** | 1.18*** | 0.76*** | 0.78*** | 1.17*** | 0.93*** |
| Power | 0.73*** | 0.69*** | 0.86*** | 0.89*** | 0.91*** | 0.83*** | 0.77*** | 0.81*** | 1.18*** | 1.09*** | 0.73*** | 0.70*** |

All analyses controlled for individual-level age and gender, as well as region-level age and sample size. See Supplementary Data 1 for the detailed results.
C individual-country fit, R individual-region fit, WB well-being, Support RS supportive relationships.
*p < 0.05, **p < 0.01, ***p < 0.001.

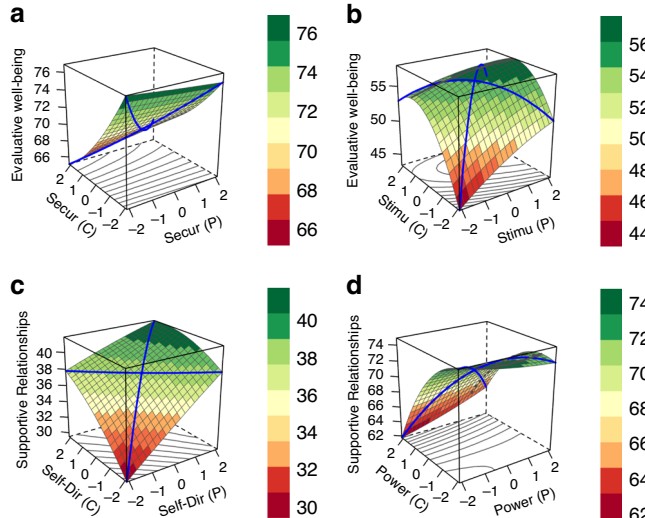

**Fig. 3 Actual associations between values and well-being.** Four response surface plots based on the multilevel polynomial regression results with individual- (P) and country- (C) level values as predictors, including control variables, for security (**a**), stimulation (**b**), self-direction (**c**), and power (**d**) values. Predictors were standardized for illustrative purposes. Source data are provided as a Source data file.

terms, the conditional $R$-squares, and the response surface coefficients can be found in Supplementary Data 1 (Online Supplemental Materials). Table 1 shows that, on most of the six dimensions of well-being, the interaction term was positive for security, achievement, and power, and negative for self-direction, stimulation, and hedonism. In contrast, fit in benevolence mattered least for well-being.

To facilitate the interpretation of the significant interactions and get a better understanding of the complex relations between values and well-being, we plotted four exemplary response surface plots (Fig. 3): two plots for positive interaction coefficients and two for negative interaction coefficients. All plots simultaneously display the two linear terms, the interaction term, and the quadratic terms (see Supplementary Data 1 for the exact coefficients). Figure 3a shows individual- and country-level security as predictors of evaluative well-being. Multilevel polynomial regression revealed significant linear effects and a significant interaction, but no quadratic effects (Supplementary Data 1). Individual security was tendentially linked with higher evaluative well-being, $B = 0.11$, SE = 0.05, $p = 0.0154$, whereas country-level security was associated with lower evaluative well-being, $B = -6.66$, SE = 1.82, $p = 0.0012$. The effect of individual-level security is difficult to spot because of the relatively large country effect. Of relevance to our hypothesis, the interaction was significant, $B = 0.64$, SE = 0.14, $p < 0.0001$, with well-being higher at the high matching points. That is, Fig. 3a shows a country-level effect, a person-level effect, and a similarity effect. The similarity effect can be mainly seen at the $x = 2|y = 2$ point, where well-being is higher than could be expected based on the two linear term effects. This interpretation is further supported by a simple slope analysis: The relation between security and evaluative well-being is absent in countries, in which people value security less on average than in countries in which people value security more, $B$s $= -0.09$ vs 0.33. We observed a similar pattern for power predicting supportive relationships (Fig. 3d).

Figure 3b shows individual- and country-level stimulation values as predictors and evaluative well-being as the outcome variable. Both the linear term of individual-level stimulation, $B = 1.02$, SE = 0.03, $p < 0.0001$, and the quadratic term, $B = -0.14$,

$SE = 0.02$, $p < 0.0001$, were significant, whereas country-level stimulation was unrelated to evaluative well-being, $B = 8.42$, SE $= 4.71$, $p = 0.0861$. Importantly, the interaction was significant, $B = -1.85$, SE = 0.17, $p < 0.0001$. Together, Fig. 3b shows linear and quadratic person-level effects, and a similarity effect. This pattern indicates that, as predicted, individuals who value stimulation report lower well-being when they live in countries in which people also value stimulation on average. This interpretation is further supported by simple slope analyses: the relation between stimulation and evaluative well-being is stronger in countries, in which stimulation is valued less on average than in countries in which stimulation is valued more, $B$s $= 1.37$ vs 0.67. We observed a similar pattern for self-direction predicting supportive relationships (Fig. 3c). Note that matches at high matching points (2|2) are not associated with lower well-being compared to (0||0) because of a strong linear effect and the absence of a quadratic effect. We also found that the pattern of results was very similar between Eastern and Western European countries (see Online Supplemental Materials).

**Individual–region fit**. Next, we tested whether individual–region fit would replicate the findings of the individual–country fit analysis reported above, using regions with at least 100 participants. This replication was obtained (Supplementary Data 2). The interaction coefficients between individual–region fit correlated strongly with those for the individual–country fit, $r(64) = 0.97$, $p < 0.001$ (mean difference between interaction terms = 0.06). Thus, operationalizing social/physical environment as a country or region does not impact the findings.

**Curvilinear relations**. In addition, we explored quadratic effects because no study has studied curvilinear relations between values and well-being, to the best of our knowledge. Across countries and regions, we found a reliable (all $p$s $\leq 0.0004$, see Supplementary Data 1 and 2) negative quadratic effect between individual-level stimulation values and all six well-being dimensions, suggesting that people who value stimulation moderately report higher well-being than those who endorse stimulation a lot or not at all. For the other values, we found no consistent pattern, although the quadratic effects tended to be negative for benevolence, universalism, self-direction, and hedonism, and positive for achievement, power, security, tradition, and conformity. Most of the quadratic effects for country-level values did not reach statistical significance.

**Discussion**
People who do not share a common frame of values with their society may feel like aliens in their own culture, but the implications of this person–society value incongruence for personal well-being have been unclear. The present research used multilevel polynomial regression to robustly examine the implications of person–society value congruence for well-being in data from over 54,000 individuals across 29 nations and 184 regions. The findings show that the effects of person–country value fit and person–region value fit depend critically on which values are examined, and in a consistent manner across dimensions of well-being.

Congruence in judgments of high importance for self-direction, stimulation, and hedonism values, relative to congruence in the midrange, was associated with lower well-being. That is, when individuals rated these values above average in importance and lived in countries or regions in which these values were also above average in importance, they reported lower well-being compared to people with matches on the average and below-average level. In contrast, congruence in achievement, power, and security values

was associated with higher well-being than congruence in the midrange. This pattern was replicated across person–region and person–country fit and demonstrated high consistency across six dimensions of well-being.

The findings support our predictions that people who highly value security experience greater well-being among other people sharing their values. They also reveal that people who highly value stimulation experience less well-being among others who share their values to the same degree. As outlined in the Introduction, a person valuing security and safety may find their need more satisfied in an environment, in which others share these values than if they do not. In addition, the opposite pattern for stimulation broadly supports the motivational opposition predicted within Schwartz's model, and it fits observations that people high in openness values identify less with their country[39] and seek to be distinct[40]. The broader findings relating to other value types can also potentially be explained by the putative underlying motivations for the values. According to Schwartz's model, self-direction, stimulation, and hedonism are anxiety-free values, whereas achievement, power, and security anxiety-avoiding values[38]. Our findings indicate that adaptation to prevalent anxiety-free values is linked to lower well-being, and adaptation to prevalent avoidance-focused values is linked to higher well-being.

It is noteworthy, however, that the patterns across dimensions of well-being were provocative but less consistent for four types of values: universalism, benevolence, tradition, and conformity. First, the pattern for other anxiety-free values, universalism, and benevolence, was less consistent across dimensions of well-being than the pattern for self-direction, stimulation, and hedonism. There are several possible explanations for this finding. Universalism and benevolence are unrelated to national identification[39] and distinctiveness motivation[40], in contrast to the other three anxiety-free values (hedonism, stimulation, and self-direction), which might have driven their effects. In addition, the results might have been restricted by ceiling effects and lower variance: in our sample, benevolence and universalism were rated as most important values, and had the smallest standard deviations, consistent with the literature[44]. One reason for the high agreement across European countries may be the presence of children in society. That is, thinking about children and being in the presence of children has been found to increase adults' prosocial values, compared to thinking about adults or being in the presence of adults[45]. In other words, children may provide persuasive arguments for the importance of self-transcendence values, such as helpfulness or protecting the environment (e.g., helping children or saving the environment for children's future). Following this argument, it would be beneficial to examine actual similarities in values in countries beyond Europe, where children may have a different societal role and presence. That is, in countries in which children, and other groups that may provide similar arguments for benevolence and universalism arguments (e.g., the elderly and disabled), are less present, might show less agreement and more variance in the importance of self-transcendence values, potentially showing stronger evidence of similarity effects in such values. Considering the importance of such groups in society, future research may test under which circumstances, and why value similarity effects occur in benevolence and universalism values. Thus, while living in benevolent egalitarian societies is beneficial to physical and psychological health, among other things[46,47], it may be the case that sharing the values that promote benevolent and egalitarian aims has indirect effects or effects that are difficult to detect with current measurement approaches.

Also, we found consistent congruence effects mainly for values with a personal focus[38]: self-direction, stimulation, hedonism,

achievement, power, and partly security. Those values are concerned with the outcome for oneself and are therefore conceptually more closely linked with indicators of personal well-being.

Second, the lack of consistency across dimensions of well-being for two avoidance-focused values, tradition, and conformity, might have arisen because congruence in values that reflect living in secure surroundings and a safe country (i.e., security) are more directly relevant to one's well-being than congruence in obedience, humbleness, and religiosity (tradition and conformity). For example, a lack of humbleness and religiosity in society may not be an as direct threat to one's well-being as a lack of secure surroundings. However, congruence in humbleness or religiosity might be relevant for personal relationships. Indeed, we only find consistent congruence effects of tradition and conformity on the well-being variable social relationships.

Interestingly, we found that the effects of individual–region and individual–country congruence were very similar. An explanation for this finding is that values were highly similar across the 29 countries investigated here, ranging between 84 and 94% (ref. [48]). This similarity may have contributed to the similar interaction coefficients between regions and countries. In other words, there is little variation in values between countries, but substantial variation within countries[49]. Nevertheless, these small mean differences in values between countries are in line with theoretical predictions and are also reliably correlated with other variables, such as level of democratization[50] and peacefulness of countries[51].

It is also conceivable that the individual–country fit is larger for people who identify more strongly with their country in general, and also more strongly with their country than the region in which they are living. Identification was not measured in this research, but identification is a relevant moderating factor in related social psychological processes[52,53]. In fact, identification is among a number of variables that may help to understand how value congruence emerges, as people might be more motivated to identify value congruence in groups with which they identify strongly[54]. Value congruence may also emerge through the same broad psychological processes that generate person situation similarity through situation perception, selection, and adaptation in the service of chronic and acute motivations[55].

Finally, a subsidiary aim of our research was to examine curvilinear effects of values on well-being. We found that stimulation is consistently associated with all six well-being dimensions in a reversed U-shape. That is, people who value stimulation somewhat also report higher well-being compared to people who value stimulation a lot or barely. This finding can be explained by the conceptual overlap of stimulation with sensation seeking. Sensation seeking can be defined as seeking intense feelings through risk taking[56]. Sensation seeking is further associated with a range of dangerous behaviors, such as high-risk sexual behavior and aggressive driving[57,58]. Conversely, too little activation can be associated with depression[59], which supports our finding that the optimum level of stimulation is not too high and not too low. This outcome was previously postulated, but not tested by Schwartz[8]: "Stimulation values derive from the presumed organismic need for variety and stimulation in order to maintain an optimal level of activation." Our findings support this reasoning, which implies that extremely high or low levels of stimulation might be less beneficial.

In summary, there are consistent effects of person–society value congruence across regions, countries, and clusters of countries (i.e., Eastern and Western Europe). In line with Schwartz's[8] influential model of human values, the nature of the relations crucially depends on which values are examined. People who highly cherish the anxiety-free values of self-direction,

stimulation, and hedonism exhibit less well-being in regions and countries, where people share their values. In contrast, people who highly cherish the anxiety-avoiding values of achievement, power, and security exhibit higher well-being if they live in regions and countries, where people share their values. This difference between the values shows that we need to rethink theoretical assumptions about simple, direct connections between value congruence and well-being. These connections are more complex and nuanced than commonly assumed. It not only matters whether the personal and national or regional values fit, but also which values fit.

## Methods

**Participants**. The 54,673 participants consisted of representative samples from 28 European countries and Israel (Albania, Belgium, Bulgaria, Switzerland, Cyprus, Czech Republic, Germany, Denmark, Estonia, Spain, Finland, France, United Kingdom, Hungary, Ireland, Israel, Iceland, Italy, Lithuania, Netherlands, Norway, Poland, Portugal, Russian Federation, Sweden, Slovenia, Slovakia, Ukraine, and Kosovo). The mean age was 48.31 years (SD = 18.59; 29,727 women, 24,929 men, and 17 who did not provide gender). These participants were obtained from the sixth wave of the European Social Survey (ESS6), which were collected in 2012 and 2013 (edition 2.3 from 1 Dec 2016). We used this wave instead of the more recent eighth wave from 2016 and 2017 for two reasons: the ESS6 contained more well-being-related variables than the ESS8, and the ESS6 included data from 29 rather than 23 countries (Ns = 54,673 vs 44,387). A sensitivity analysis revealed that the sample size allowed us to detect effects as small as r = 0.01 at a power of 0.95. The research was approved by the ERIC Research Ethics Board of the European Social Survey (https://www.europeansocialsurvey.org/about/ethics.html).

The 29 countries were further divided into 351 regions, such as 12 regions in the UK (9 English regions, and Northern Ireland, Scotland, and Wales) or the 16 federal states in Germany (e.g., Bavaria, Saxony, and Berlin). To ensure sufficiently large sample sizes, we only included respondents from regions with at least 100 participants, leaving 184 regions with 45,282 participants. Israel (n = 2508) and Cyprus (n = 1116) were treated as one region each (i.e., participants in those two countries were not split up into regions). The smallest included regions (all ns = 100) were Pardubice (Czech Republic), Pomeranian Voivodeship (Poland), Kiev Oblast (Ukraine), Pays de la Loire (France), and Bretagne (France).

**Measures**. Values were measured with the 21-item version of the Portrait Value Questionnaire[60,61]. The Portrait Value Questionnaire measures the ten value types in Schwartz's[8] model. Using a scale from 1 (very much like me) to 6 (not like me at all), participants indicated how similar they were to a fictitious person who exemplifies one of the ten value types. Examples for items include "S/he thinks it is important that every person in the world be treated equally. S/he wants justice for everybody, even for people s/he doesn't" (universalism) and "It is important to her/him to be rich. She/he wants to have a lot of money and expensive things" (power). Prior to performing any analysis, we recoded all 21-items so that higher scores indicate greater endorsement. Previous research established configural and metric invariance, suggesting that associations between values can be meaningfully compared across countries[62].

We measured well-being with the six dimensions identified by Jeffrey, Abdallah, and Quick[37]. Evaluative well-being was measured with two items that asked how satisfied and happy participants were (r = 0.71). Emotional well-being was measured with six items that asked how often participants felt various emotions, such as depressed (recoded), anxious (recoded), and peaceful (α = 0.82). The 14-item functioning scale asked whether participants were free to decide how to live their life, were enthusiastic about what they are doing, or felt like a failure (recoded; α = 0.85). Thus, functioning is closely related to self-esteem[37]. Using four items, the vitality scale asked whether participants had sleep issues (recoded) or had a lot of energy in the past week (α = 0.69). The five-item community well-being scale asked participants how close they feel to other people and whether they can trust them (α = 0.67). This scale contained the three items that Beilmann and Lilleoja[19] used to measure social trust. The four-item supportive relationship scale asked respondents how often they felt lonely in the past week (recoded) or appreciated by those to whom they are close (α = 0.57). The response scales differed across the well-being dimensions. Items were standardized before they were averaged to form the dimensions. The six well-being dimensions were positively correlated with each other (median r = 0.52). However, the measure is likely not fully invariant across countries: Charalampi et al.[63] analyzed the six-dimensional well-being measure in 17 out of the 29 countries that participated in the sixth round of the European Social Survey. They found the best statistical fit for a four-factor solution in some countries, a five-factor solution in other countries, and a six-factor solution in only two countries. However, we do not expect different results if the items were combined into different factors, because the six dimensions correlated highly with each other, on average (median r = 0.52). Further, it is typically difficult to establish measurement invariance across a larger set of items, factors, and especially countries. Indeed, Charalampi et al. concluded that "the analysis did produce

reliable and valid summary measures (subscales) of well-being for informing social policy in each country" (p. 73).

Following Bleidorn et al.[29], we T-transformed the six dependent variables (standard scores with M = 50 and SD = 10), because T-scores can be used as an effect size where a two-point difference represents a small effect, a five-point difference a medium effect, and an eight-point difference a large effect.

**Data analysis**. We used multilevel polynomial regression analysis and response surface plots[26,28,64] to test whether value congruence is associated with well-being. We controlled for country- or regions-level sample size, following Bleidorn et al.[29]. The pattern of results were similar when we controlled for individual-level age and gender, as well as country-level age: correlations between linear terms of individual-level values predicting well-being without and with controlling for age and gender were r(58) = 0.97, and correlations between the interaction terms without and with controlling for age and gender were r(58) = 0.99 (Supplementary 3 and 4). Also, controlling for the country-level variable Human Developmental Index (HDI)[65], individual-level education, or income resulted in the same pattern of findings (Supplementary Data 5 and 6). For example, the correlation between the interaction terms without control variables as reported in Supplementary Data 1, column 9, and the interaction terms of the multilevel polynomial regressions in which we controlled for individual-level age, gender, education level, income, country-level age, and the HDI was r(58) = 0.98. A paired-sample t test comparing the interaction terms was not significant, t(59) = 0.82, p = 0.42, suggesting that the effects we found are robust. However, particularly including the HDI as a country-level control variable reduced; however, the country-level effects substantially (Supplementary Data 5). These are less relevant for the presented research (the reduced country-level effects can be explained by previous research suggesting strong associations between country-level scores of well-being and other country-level variables, such as income)[41]. The finding that HDI does not affect the congruence effects is echoed by our supplemental analysis, where we found that the pattern of results was very similar across Eastern and Western European countries (see Online Supplemental Materials).

Following recommendations, predictors were centered along their respective mean[28,66] before they were multiplied to form the interaction term and squared to form the quadratic terms. In addition, we verified the occurrence of matches and mismatches, and that predictors are not multicollinear.

In total, we performed 10 (value types) × 6 (well-being dimensions) × 2 (country vs region) polynomial regressions to examine the effects of fit. Because of our large sample size and high number of comparisons, we will only discuss findings that are significant at α = 0.001. We consider this threshold as neither overly conservative nor liberal. However, readers might vary in their preferred thresholds, and we therefore report the exact p values up to four decimal places in the Online Supplemental Materials. To interpret the results, we focus on the regression weights separately[32]. Some researchers argued that to investigate the interaction, as well as linear and quadratic effects further, one should compute the four response surface parameters, a1–a4, which allow for a more detailed interpretation of the response surface plots[28,64]. For example, the parameter a4 indicates whether well-being is higher (positive a4) or lower (negative a4) in the presence of large mismatches between own and regional-level values. Unlike the interaction term (Fig. 2), the outcome (here: well-being) for all matches is the same, no matter whether a4 is positive or negative.

For a similarity effect to occur, Humberg et al.[64] argued that first the coefficient a4 needs to be significantly negative. However, we believe this expectation is problematic because of the way that a4 is computed: it is the sum of both quadratic terms minus the interaction term[26,28]. Thus, the quadratic effects are weighted twice as heavily as the interaction term in the computation of a4, masking congruence effects at the extreme ends of the predictors, as indicated by the interaction term. Hence by looking at the interaction term and the quadratic terms separately, we can obtain more detailed information about any congruence[32].

Following Bleidorn et al.[29], we did not formally test for congruence. Specifically, Edwards[26,64] argued that, among other things, a strict requirement for congruence is that the ridge of the plane needs to be exactly above the main diagonal of the plot, that is between the points (−2|−2|−4) and points (2|2|−4), as it is the case for both planes displayed in Fig. 2. This requirement is difficult to achieve in the present research because the variance for the country-level scores is significantly smaller than for the individual-level value scores. To estimate the amount of explained variance, we computed the conditional $R^2_{GLMM}$[67] with the R-package MuMIn[68] (version 1.42.1), which includes fixed and random effects. Multilevel analyses were conducted with the R-package lme4 (ref. [69]; version 1.1-18-1), as well as response surface analyses with the R-packages RSA[70] (version 0.9.12) and ggplot2 (ref. [71]) (version 3.3.0).

Some value researchers recommend to ipsatise (or center) the value scores on an individual level by subtracting the mean score of each participant from their value rating[8,60]. It is argued that ipsatising corrects for individual scale use, and allows researchers to investigate the relative importance people attribute to each value rather than the absolute importance. However, we did not ipsatise the value scores for several reasons. First, He and van de Vijver[72] challenged empirically the claim that ipsatising ("centering") corrects for scale use: "[s]core corrections to deal with response styles are not recommended" (p. 129). Second, He et al.[73] found that ipsatisation reduces internal consistency and measurement invariance, which is

especially important for cross-cultural research (note that Davidov et al.[62], cited above, do not report to have ipsatised the value scores prior to testing for measurement invariance across countries). Third, we already centered the data on a variable level as recommended by guidelines for moderated regression analysis[66]. Ipsatising and then centring the data would reduce the interpretability of the data. Fourth, Borg and Bardi[74] reported that the mean rating of the value responses of each person is correlated with well-being. That is, how much participants rated the importance of on average with all value items correlated positively with well-being. For example, people who are depressed are less likely to find any value important. Reproducing the approach of Borg and Bardi in the dataset, we are using yielded comparable correlations between the mean ratings and the six well-being dimensions of $r = 0.04$ (communal well-being) and $r = 0.33$ (functioning). Thus, if we were to use the difference between value scores and the mean rating, which is itself is correlated with well-being, findings would be distorted. Finally, researchers who work with conceptually related constructs, such as personality traits (e.g., Big Five) or goals do not ipsatise, which would hamper cross-construct comparisons.

**Reporting summary**. Further information on research design is available in the Nature Research Reporting Summary linked to this article.

## Data availability
The data is openly available (http://www.europeansocialsurvey.org/). A reporting summary for this article is available as a Supplementary information file.

## Code availability
The R-code for the analyses can be found on https://osf.io/u6378.

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

## Acknowledgements

This project was partly supported by funding from the Economic and Social Research Council (ESRC) under grant agreement ES/P002463/1. The funders had no role in study design, data collection and analysis, decision to publish, or preparation of the manuscript.

## Author contributions

Conceptualization: P.H.P.H., U.W., L.J.W., G.L.d.H.C., and G.R.M.; data curation: P.H.P.H.; formal analysis: P.H.P.H.; methodology and validation: P.H.P.H., U.W., L.J.W., G.L.d.H.C., and G.R.M.; visualization: P.H.P.H. and L.J.W.; and writing: P.H.P.H., U.W., L.J.W., G.L.d.H.C., and G.R.M.

## Competing interests

The authors declare no competing interests.
