## [Peer Review File · Nature Communications]

Reviewers' comments:

Reviewer #1 (Remarks to the Author):

The paper "Well-being as a function of person-country fit in human values" reports on a very interesting endeavor, namely to explore to what degree the fit of individual-level value preferences with country-wide value preferences relates to (several different aspects of) well-being.

The analytical strategy pursued in the paper is outright excellent. Having said that a meta-critical question does come to my mind. Shalom Schwartz grounded his most convincing substantive arguments in the simplest analytical strategies: He presented bivariate correlations and (SSA) plots thereof. When I read about most sophisticated analytical approaches (as the ones I am confronted with in the present article), I sometimes cannot avoid thinking that the presented findings only emerged because hyper-sophisticated analytical methods were used. This aspect of possibly 'overpowering' one's analyses is rarely if ever discussed in the methods literature. Allow me an excursus to exemplify what I mean. Let us assume we analyze panel (aka longitudinal) data on value development across the lifespan. In the 'olden' times researchers would have analyzed such data using repeated measures ANOVA. If they had very many measurement points, ARIMA models were an alternative. Nowadays, latent growth modelling is a must. Mere ANOVAs would not get published anymore in high impact journals. But couldn't one also argue that effects that emerge only in LGM analyses, but not in Repeated Measures ANOVAs, are overstraining the psychological substance as obtained through a survey methodology? Bringing it back to the current paper: I would have loved to see whether results also emerge if analyses are based on distances (difference scores of whatever algorithmic operationalization), knowing that many objections have been brought forward against them. I do believe, though, that distances are closer to what really happens in people's cognitions (than the approach utilized in the paper).

This brings me to my other main concern. Rudely formulated (I apologize), the paper takes the approach of poking in some (admittedly hyper-interesting) fog. The question of what is the psychological mechanism behind a positive relationship of person-country value congruence and well-being remains grossly underdeveloped in the paper. WHY should my well-being gain (or lose) from congruence with a fictitious entity, namely a country-specific value preference profile, and HOW do country-wide value preferences affect people's cognitions? In my view, the paper's argumentation lacks strength in this respect. Adding conceptual elaborations is urgently necessary. How do people relate to the country-wide value preferences? Staying immanently in Schwartz's value theory might work: People who prioritize conservation values tend to be against 'being different.' So for them, experiencing that the people around them have similar preferences is soothing, thereby enhancing their well-being. For people who prioritize openness values, values of being different from others in many ways, getting to know that they are by no means different from their fellow countrymen and women, may be disappointing and thereby impeding their well-being. The sketched possibility is, however, not clearly discussed in the paper, which does need a thorough revision in this regard.

What is furthermore evident when reading the paper is the fact that it argues almost exclusively intra-

psychologically. But, how about societal processes? In value transmission studies, it has been shown several times that value similarity between parents and off-spring is strongest for values that are UNpopular in a given society. Could culture-specific relative popularity of a given value in a given society maybe be a moderator of the value-congruence-well-being relationship?

In summary: Highly interesting topic, high methodological sophistication--but maybe oversophistication and a--sorry--shallow conceptual argumentation as to why the presumed relationship exists and how it functions.

Reviewer #2 (Remarks to the Author):

This study investigated whether well-being is higher if people's values match those of people living in the same country or region. The study examines the objective match between people's own values and the measured values that prevail in their country and region rather than the values people perceive to prevail. The study examines values in representative national samples from 29 countries. There is a substantial literature on relations with well-being of person-environment fit in values. The current research goes beyond this literature, innovating in three important ways.

First, the study examines relations of fit to six different aspects of well-being rather than to the one or two aspects typical of other studies. Consistent results across the indexes of well-being greatly reinforce our confidence in the findings.

Second, the study replicates its main findings for value congruence both with country values and with region values. This further reinforces confidence.

Third, rather than using a global index of value fit, the study examines the relations to fit separately for each one of the ten values in the Schwartz (1992) value theory. This leads to the most novel and exciting finding of the study that will be of great interest in the field: The effects of value fit depend upon the particular value. For those who cherish openness values, value fit is detrimental to their well-being whereas for those who cherish self-enhancement and security values, value fit is beneficial. The authors provide interesting theorizing to explain these findings.

Moreover, the methods employed to assess fit are more sophisticated and informative than in most previous studies, thereby overcoming flaws common to many of these studies. These methods and their visual representations enable us to distinguish between the linear effects of individual values on well-being, the linear effects of country values, and the effects of interactions between them, which is a pure index of value fit/match.

Below, I offer numerous comments that I hope will contribute to improving what is already a fine manuscript.

1. Abstract: The sentences describing the findings for openness and self-enhancement values are complex and difficult to parse. Perhaps they can be made easier to grasp.
2. P3: Need a citation for the statement that the circular pattern of values has been confirmed in over 80 countries.
3. Figure 1 caption: Better to describe the model as circular (or quasi-circumplex), not as a circumplex.
4. P5: Why say that differences in statistical approaches may have the strongest impact on the

inconsistencies in the literature? Any justification for speculating about the strength of impact?

5. P8: The prediction for the effect of congruence on those who value conservation might be based on the idea that a congruent value environment is important to provide what those who value conservation are seeking in life—security, stability, social support, sense of belonging. These outcomes are especially important to those who value security, less so for tradition and conformity, which may account for why only security confirmed the hypothesis.

6. P10: Although the main focus of the paper is on congruence effects, if space allows, adding a figure for either security or stimulation that shows the linear effects in addition to the interaction would help readers to understand.

7. P11: Might a comment on the measurement invariance of the value indexes be desirable?

8. P13: I had difficulty understanding the last two sentences of the second paragraph.

9. P14: last line before table: Benevolence mattered least, but tradition was about the same as conformity.

10. P19: You offer a number of explanations for why self-transcendence values did not show the same consistent patterns as openness values with which they share the designation 'anxiety-free'. Self-transcendence values differ from openness values in their social vs. personal focus. This may provide another key to understanding the different results. For example, satisfying self-transcendence values involves reaching out to others in a supportive, closeness-building way. This may conflict with seeking to be distinctive. The social vs. personal focus may also be relevant for understanding the findings for the anxiety avoidance values. Security values have a mixed social and personal focus, conformity and tradition do not.

11. P20: The first two sentences are confusing. Do you mean values in regions were similar to those of their country? Please clarify. If you mean little variation in values across countries, how could there be meaningful differences in country level values.

12. P20: The curvilinear association of stimulation values with well-being is implied in the original presentation of this value in Schwartz, 1992: "Stimulation values derive from the presumed organismic need for variety and stimulation in order to maintain an optimal level of activation (Berlyne, 1960; Houston & Mednick, 1963; Maddi, 1961)." 'Optimal' implied that the need is not for an extremely high or low level. Your novel finding supports this idea.

13. P21: The fact that achievement values behaved like anxiety avoiding values in this study is interesting. They are at the border of self-enhancement and openness in the circle. I defined them as seeking approval for success, but there seems to be a tendency of some respondents to understand them as more intrinsic pursuit of successful performance. This may be less the case with the ESS items than with the PVQ versions. Another possible explanation for the congruence effect for power and achievement may be that behavior in pursuit of these two values is liable to elicit negative reactions from others because it is counter-normative. But in countries where these values are widely cherished, such behavior is more normative, so negative reactions are weaker.

14. Why are all the simple linear effects of values positive? This is contrary to literature summarized in Schwartz & Sortheix (2017) that shows fairly consistent negative effects of tradition, conformity, security and power? This deserves some comment.

15. It would be helpful to add to the supplement some explanation of the multilevel analysis used and more on the polynomial method, so readers do not have to search sources if they wish to understand in

more depth.
Shalom Schwartz

Reviewer #3 (Remarks to the Author):

This manuscript examines whether objective value similarity at the regional and country-level relate to individuals' well-being. The paper draws on the European Social Survey vast dataset, including over 50.000 individuals, and uses a methodology that has not been applied to analyse this issue: multilevel polynomial regression and surface response analysis.

The main findings are that people who highly value self-direction, stimulation, and hedonism, and who live in countries and regions where these values are also highly valued on average report lower well-being than people who ascribe lower importance to these values and live in countries and regions where people on average give low importance to these values. In this case dissimilarity increases WB. We found the reversed pattern for achievement, power, and security values. That is individuals who value these values highly are more satisfied when they live in regions/countries where these values are also highly valued.

In summary, the main problems I find are with the centering procedure for values and lack of country-level controls for socio-economic development. Also, it is unclear the centering used for testing the cross-level interactions and how this would influence the interpretations. Furthermore, I'm missing the mechanisms that would explain your findings.

On a minor note: The expression that they live in places where "people share their values" in the abstract is misleading because the authors acknowledge that within-country variability of values is very high. Thus, we cannot interpret that those who value OP highly necessarily interact with others who also value it highly.

Introduction

1. The authors formulate expectations for openness to change values at the individual and country-level, which would be positive for well-being. But why are there no expectations for the conservation values? In past studies using large representative samples those have been consistently linked to lower well-being (e.g., Bobowik et al., 2010; Sortheix & Lönnqvist, 2014, or Sortheix & Schwartz, 2017). Also collectivism at the country level correlates with lower SWB. So, I wonder why only openness was mentioned here?

2. It is also not clear why they expected that only value congruence in conservative values would predict well-being? As the authors wrote one could also expect similarity to be beneficial for all values, similarly to personality traits findings (another reference: Fulmer, C. A., Gelfand, M. J., Kruglanski, A. W., Kim-Prieto, C., Diener, E., Pierro, A., & Higgins, E. T. (2010). On "feeling right" in cultural contexts: How person-culture match affects self-esteem and subjective well-being. *Psychological Science*, 21(11), 1563-1569.]

Methods

3. The authors have excellent methodological knowledge and cite all appropriate references. However,

some comments. In the description of the value measure, the PVQ21, the authors do not tell if they reversed the items so that higher scores would imply higher endorsement of the value (I assume they did this).

4. More importantly, the authors don't tell whether they corrected for scale use by centering participants' responses around their own mean, which converts values into priorities. I think it is a serious problem if the authors did not do this. I assume they didn't, as all values have positive coefficients with the outcomes instead of following a pattern of negative to positive correlation as the expected sinusoid curve would anticipate.

5. In the results, this could explain that security is "positively" related to well-being (page 15) even though it has the smallest positive coefficient of all values (Annex) and if values would be centered as in most research using the Schwartz, I assume security would be negative. This is even more relevant if we want to compare these results with past studies using the same ESS dataset and showing security was negatively related to WB (e.g., Bobowik et al., 2010; Sortheix & Lönnqvist, 2014, or Sortheix & Schwartz, 2017). Those studies used centered value scores and results (although not perfectly) follow the expected sinusoid curve: security, power were negatively associated with well-being while hedonism and benevolence were positively related, in general (life satisfaction, negative, positive emotions).

6. Ideally, the cross-cultural equivalence for scales for well-being should have been assessed, but I think that with such a large number of items this is difficult. It should at least be mentioned in limitations. Particularly, I wonder what this scale with 14 items really measures and if there is any validity for it: "The 14-item functioning scale asked whether participants were free to decide how to live their life, were enthusiastic about what they are doing, or felt like a failure (recoded; $\alpha = .85$). I know of one study that found very limited validity by the measurement proposed by Jeffrey et al. (2015) that you used. [Charalampi, A., Michalopoulou, C., & Richardson, C. (2018). Validation of the 2012 European Social Survey Measurement of Wellbeing in Seventeen European Countries. *Applied Research in Quality of Life*, 1-33.]

7. On page 14, please, open up for the reader what you mean by: "Following Bleidorn et al. (2016), we did not test formally for congruence (Humberg et al., 2018)".

8. The centering procedure for all variables is not sufficiently explained. Did you grand or group-mean centered?

Results

9. Page 15: The relation between security and evaluative well-being is absent in countries in which people value security less on average than in countries in which security is more valued, $B_s = 0.00$ vs 0.43 . " Does the interpretation also imply that when living in countries where security is less valued they report lower well-being? If we think about this in real life, this would mean that a person who values highly security (which as I wrote before predicts lower WB), and lives in Poland or Italy (high on security in ESS), he/she is more satisfied than if she/he lives in Finland (relatively low on security)? Those countries high on security are generally characterized by their instability, making life more insecure (contradicting the goals of conservative values), as we know that more socio-economic development correlates with higher individualism. That argument was used by past research to argue why the relation of security and other conservation values was more strongly negative in countries with lower HDI (Sortheix & Lönnqvist, 2014) or less egalitarian (Sortheix & Schwartz, 2017). However, I fail to see what mechanism would explain your findings. Have you tested if your results are robust when you control for

socio-economic development?

10. We observed a similar pattern for power predicting supportive relationships” And for the other outcomes? The interaction was significant for all outcomes. So, I’m not sure why you mention only this one.

11. How do you explain your findings for openness? Your results would mean that a person who highly values openness and lives in Denmark (high in OP) or Copenhagen region (high in OP) would report lower well-being.... This doesn’t make sense to me. The interpretation in the discussion also goes against past research showing that people enjoy higher well-being in anxiety-free contexts, e.g., more autonomous, self-expression societies (see Schwartz, Inglehart and Welzel model of human development, Fischer and Boer’s findings). So, I find this sentence ungrounded: “Our findings indicate that adaptation to prevalent anxiety-free values is linked to lower well-being, and adaptation to prevalent avoidance-focused values is linked to higher well-being. “ It would be then important to show whether results hold after including HDI or GDP and offer a more convincing explanation.

12. In my view, for this paper to have more influence in the field, it would be still needed /useful to compare and offer explanations that enlighten other studies using the same ESS dataset which also found dissimilarity effects using other indicators at the country-level. For instance, Sorthaix and Lönnqvist (2014) used HDI and found that individuals high in OP values who live in low HDI countries (which are generally lower on OP at the country-level) report higher life satisfaction. It would be necessary to find a novel argument for what you found.

Reviewer 1

The paper "Well-being as a function of person-country fit in human values" reports on a very interesting endeavor, namely to explore to what degree the fit of individual-level value preferences with country-wide value preferences relates to (several different aspects of) well-being.

The analytical strategy pursued in the paper is outright excellent. Having said that a meta-critical question does come to my mind. Shalom Schwartz grounded his most convincing substantive arguments in the simplest analytical strategies: He presented bivariate correlations and (SSA) plots thereof. When I read about most sophisticated analytical approaches (as the ones I am confronted with in the present article), I sometimes cannot avoid thinking that the presented findings only emerged because hyper-sophisticated analytical methods were used. This aspect of possibly 'overpowering' one's analyses is rarely if ever discussed in the methods literature. Allow me an excursus to exemplify what I mean. Let us assume we analyze panel (aka longitudinal) data on value development across the lifespan. In the 'olden' times researchers would have analyzed such data using repeated measures ANOVA. If they had very many measurement points, ARIMA models were an alternative. Nowadays, latent growth modelling is a must. Mere ANOVAs would not get published anymore in high impact journals. But couldn't one also argue that effects that emerge only in LGM analyses, but not in Repeated Measures ANOVAs, are overstraining the psychological substance as obtained through a survey methodology? Bringing it back to the current paper: I would have loved to see whether results also emerge if analyses are based on distances (difference scores of whatever algorithmic operationalization), knowing that many objections have been brought forward against them. I do believe, though, that distances are closer to what really happens in people's cognitions (than the approach utilized in the paper).

We agree with the Reviewer that more advanced analytical methods can come with a cost of reduced clarity. However, we believe that the polynomial regression analysis in combination with the response surface analysis improves clarity over difference scores and profile correlations. As Edwards (2002) notes, “[D]ifference scores are often less reliable than either of their component measures. Difference scores are also inherently ambiguous, given that they combine measures of conceptually distinct constructs into a single score. Furthermore, they confound the effects of their component measures on outcomes and impose constraints on these effects that are rarely tested empirically. Finally, they reduce an inherently three-

dimensional relationship between their component measures and the outcome to two dimensions.” (p. 351). Therefore, “due to the problems with difference scores, results from studies that use difference scores are uninterpretable and should be discounted” (Jeffrey Edwards, personal correspondence).

In contrast, response surface analysis also relies on differences, but on more sophisticated ones than the classical difference scores. For example, it allows researchers to compare participants who score high on both predictors with those who score low on both predictors. This would not be possible with absolute difference scores. Take two hypothetical participants as an example. Participant A scores high on a given value (i.e., response option “6”) and lives in country X where people on average share the value to the same extent (i.e., “6”). Participant B scores low on a given value (i.e., response option “1”) and lives in country Y where people on average share the value to the same extent (i.e., “1”). The (absolute) difference scores for both participants are the same: $6 - 6 = 1 - 1 = 0$. We hope that this example demonstrates why polynomial regression and response surface analysis are clearer than (absolute) difference scores. Further, polynomial regression and response surface analysis also provide plots which further illustrate the nature of the findings in ways that are not possible with a difference score or correlation approach. To clarify this, we now added a more detailed description of the response surface analysis to the Introduction “For example, the complex interplay between the two predictors can be visually displayed using response surface analysis. This plot displays a surface with the estimated outcome of all possible combinations of the predictors. Such plots allow researchers to easily identify the location of the similarity effect (e.g., is well-being higher when both a person and other people same in the country they are living in value security above average?).” (p. 6)

Nevertheless, to comply with the Reviewer’s request, we correlated the difference scores and absolute difference scores (participants’ values minus country averages) with the dependent variables (120 correlations in total) and report the findings in the Supplemental Materials. The R-code to reproduce the analysis is available at the end of the R-script which we uploaded on https://osf.io/u6378/?view_only=1bc30284f50645e390bb021c45fa3628 However, for the reasons stated above we do not mention this analysis in the main document.

This brings me to my other main concern. Rudely formulated (I apologize), the paper takes the approach of poking in some (admittedly hyper-interesting) fog. The question of what is the psychological mechanism behind a positive relationship of person-country value congruence and well-being remains grossly underdeveloped in the paper. WHY should my well-being gain

(or lose) from congruence with a fictitious entity, namely a country-specific value preference profile, and HOW do country-wide value preferences affect people's cognitions? In my view, the paper's argumentation lacks strength in this respect. Adding conceptual elaborations is urgently necessary. How do people relate to the country-wide value preferences? Staying immanently in Schwartz's value theory might work: People who prioritize conservation values tend to be against 'being different.' So for them, experiencing that the people around them have similar preferences is soothing, thereby enhancing their well-being. For people who prioritize openness values, values of being different from others in many ways, getting to know that they are by no means different from their fellow countrymen and women, may be disappointing and thereby impeding their well-being. The sketched possibility is, however, not clearly discussed in the paper, which does need a thorough revision in this regard.

In the previous version of the manuscript, we had alluded to the points suggested by the reviewer. For example, we wrote “As a starting point, we predicted that people with high conservation values (e.g., security, tradition, and conformity), which preserve the status quo, would report higher well-being when these values are shared by others in their country because this will make their life more stable. Also, people valuing conservation values identify more with their country (Roccas et al., 2010) and prefer to be similar to people around them (Eriksson et al., 2011), which likely results in higher well-being.” (p. 9) while referring to the need to be distinct when predicting that this pattern is likely reversed for people high in openness values. We returned to these explanations in the Discussion.

However, we take the reviewer’s point that our literature review on the person-environment fit was somewhat underdeveloped and have now expanded it by linking it more strongly to the literature. We now write “The values described in this model can be examined at the level of individuals and aggregate groups, such as countries, and congruence between personal and national values has theoretical implications for personal well-being. Four potential reasons for this link are environmental affordances, social sanctions, internal conflict, and shared realities (cf. Edwards & Cable, 2009; Fulmer et al., 2010; Higgins, 2019; Sagiv & Schwartz, 2000; Solomon & Knafo-Noam, 2007; Stromberg & Boehnke, 2001). For example, if an individual’s society values achievement (e.g., success), and the individual does as well, then the individual might feel supported by the sense of common purpose. If the individual does not value achievement, the individual might feel pressured by social norms to achieve more. This pressure, in turn, may reduce the individual’s subjective well-being.

Further, incongruence between people's own and their compatriots' values can result in internal conflicts, which may result in reduced well-being. This reduction might occur because people's goals are blocked in incongruent environments. Conversely, goal facilitation may occur in congruent value environments. Moreover, congruence in values can result in the experience of shared reality and identity, which in turn helps to validate one's experience, facilitates co-operation, coordination, and promotes a sense of belonging (Higgins, 2019; Orbell et al., 1988)." (p. 4f)

What is furthermore evident when reading the paper is the fact that it argues almost exclusively intra-psychologically. But, how about societal processes? In value transmission studies, it has been shown several times that value similarity between parents and off-spring is strongest for values that are UNpopular in a given society. Could culture-specific relative popularity of a given value in a given society maybe be a moderator of the value-congruence-well-being relationship?

We thank the reviewer for this suggestion. This actually serves as another example of the utility of the polynomial approach, which we have tested by correlating the absolute scores of the interaction terms of the polynomial regressions with the 10 means of the country-level (and separately with region-level) value types. If the reviewer's prediction is correct, the interaction terms should be weaker (e.g., closer towards 0) if the values are rated as more important. However, the correlation between the individual-country interaction and the means of the 10 value types was non-significant, $r(58) = -.15, p = .27$, as was the correlation between the individual-region interaction and the means of the 10 value types, $r(64) = -.18, p = .17$ (the correlations using the raw numerical interaction scores as opposed to the absolute scores were weaker, $r_s = -.07$ and $-.10$). We then re-read value transmission studies and found that the claim put forward by the reviewer was only supported, in our reading, by Boehnke (2001), but neither by Yi et al. (2004) nor Roest et al. (2009). We therefore decided not to discuss this additional analysis in the manuscript, but are not resistant to the possibility if you or the reviewer prefer us to include it.

Furthermore, we followed the suggestions of Reviewer 3 and investigated other societal processes such as education level or income (see below).

Reviewer 2

This study investigated whether well-being is higher if people's values match those of people living in the same country or region. The study examines the objective match between people's own values and the measured values that prevail in their country and region rather than the values people perceive to prevail. The study examines values in representative national samples from 29 countries. There is a substantial literature on relations with well-being of person-environment fit in values. The current research goes beyond this literature, innovating in three important ways.

First, the study examines relations of fit to six different aspects of well-being rather than to the one or two aspects typical of other studies. Consistent results across the indexes of well-being greatly reinforce our confidence in the findings.

Second, the study replicates its main findings for value congruence both with country values and with region values. This further reinforces confidence.

Third, rather than using a global index of value fit, the study examines the relations to fit separately for each one of the ten values in the Schwartz (1992) value theory. This leads to the most novel and exciting finding of the study that will be of great interest in the field: The effects of value fit depend upon the particular value. For those who cherish openness values, value fit is detrimental to their well-being whereas for those who cherish self-enhancement and security values, value fit is beneficial. The authors provide interesting theorizing to explain these findings.

Moreover, the methods employed to assess fit are more sophisticated and informative than in most previous studies, thereby overcoming flaws common to many of these studies. These methods and their visual representations enable us to distinguish between the linear effects of individual values on well-being, the linear effects of country values, and the effects of interactions between them, which is a pure index of value fit/match.

We thank the reviewer for this positive and thoughtful evaluation of our paper.

Below, I offer numerous comments that I hope will contribute to improving what is already a fine manuscript.

1. Abstract: The sentences describing the findings for openness and self-enhancement values are complex and difficult to parse. Perhaps they can be made easier to grasp.

After re-reading again, we take the point and have revised the sentences. The original sentence read “People who value self-direction, stimulation, and hedonism and live in countries and regions where people share their values report lower well-being than people who ascribe lower importance to these values and live in countries and regions where people endorse the values to a similar extent. We found the reversed pattern for achievement, power, and security values.” The revised sentences now read “People who value self-direction, stimulation, and hedonism more and live in countries and regions where people on average share these values reported *lower* well-being. In contrast, people who value achievement, power, and security more and live in countries and regions where people on average share these values, report *higher* well-being.”

2. P3: *Need a citation for the statement that the circular pattern of values has been confirmed in over 80 countries.*

We now added “(Schwartz, 2018)” as reference at the end of the sentence on page 3.

3. *Figure 1 caption: Better to describe the model as circular (or quasi-circumplex), not as a circumplex.*

We now write “quasi-circumplex”.

4. P5: *Why say that differences in statistical approaches may have the strongest impact on the inconsistencies in the literature? Any justification for speculating about the strength of impact?*

We acknowledge that this point is somewhat speculative (hence “*may* have the strongest impact” [emphasis added]). Nevertheless, we believe that other differences between the studies are probably less relevant for explaining differences between studies. For example, differences between the objective and subjective approach (i.e., whether studies tested actual vs perceived value similarities) are according to a meta-analysis from Montoya et al. (2008) on personality traits and attitudes similarity negligible. Specifically, the effect sizes were very similar (.54 vs .49) when no direct interaction between individuals has taken place, which is the case for most studies regarding value congruence that we are reviewing in the manuscript. A third difference between the studies on effects of value congruence is the type of sample:

students vs non-students. While there is some evidence that effects obtained from students do not necessarily replicate in non-student samples (e.g., Hanel & Vione, 2016; Peterson, 2001), in human value research the pattern of results is often similar. For example, the hierarchy of values between student samples and representative samples is similar (Schwartz & Bardi, 2001) and how people perceive the values of other people is similar across sample types (Hanel et al., 2018). However, again, we acknowledge that this is speculative and may confuse readers. We therefore now write “Further, the statistical approaches differ, which may also have a strong impact on the outcome.” (p. 5).

5. P8: The prediction for the effect of congruence on those who value conservation might be based on the idea that a congruent value environment is important to provide what those who value conservation are seeking in life—security, stability, social support, sense of belonging. These outcomes are especially important to those who value security, less so for tradition and conformity, which may account for why only security confirmed the hypothesis.

We thank the Reviewer for this suggestion, which adds to our original thinking, and therefore we have included it as an additional possibility in the Discussion (p. 19f): “the lack of consistency across dimensions of well-being for two avoidance-focused values, tradition and conformity, might have arisen because congruence in values that reflect living in secure surroundings and a safe country (i.e., security) are more directly relevant to one’s well-being than congruence in obedience, humbleness, and religiosity (tradition and conformity). For example, a lack of humbleness and religiosity in society may not be an as direct threat to one’s well-being as a lack of secure surroundings. However, congruence in humbleness or religiosity might be relevant for personal relationships. Indeed, we only find consistent congruence effects of tradition and conformity on the well-being variable social relationships.” (p. 21) The revised paper does not discuss social support or sense of belonging as security values, because they are not measured by the PVQ-21 items.

6. P10: Although the main focus of the paper is on congruence effects, if space allows, adding a figure for either security or stimulation that shows the linear effects in addition to the interaction would help readers to understand.

The three-dimensional plots (Figures 2-3) already contain all five terms, which we now clarify on page 16: “All plots simultaneously display the two linear terms, the interaction term, and the quadratic terms (see Table S1 for the exact coefficients).” We agree that it is

significantly easier to spot a linear effect in a two-dimensional than a three-dimensional plot. We have not added regression lines to display the linear and quadratic effects because we are worried that this would be overwhelming. We are hoping that the interested reader will find the detailed results reported in the supplemental materials sufficient.

7. P11: Might a comment on the measurement invariance of the value indexes be desirable?

We have now added to the “Material” subsection: “Previous research established configural and metric invariance, suggesting that associations between values can be meaningfully compared across countries (Davidov et al., 2008).” (p. 12)

8. P13: I had difficulty understanding the last two sentences of the second paragraph.

Those two sentences discuss problems interpreting the response surface parameter a_4 . Recall that the parameter a_4 reveals whether well-being is higher (positive a_4) or lower (negative a_4) in the presence of large mismatches between own and regional-level values. The a_4 is the sum of both quadratic terms minus the interaction term and is thus a composite score, making it more informative to interpret the interaction term and the quadratic terms separately. We unnecessarily belabored this point in our submitted manuscript, given that we nonetheless report a_4 in the supplemental materials alongside the other terms. We therefore have now moved the two paragraphs into a footnote (Footnote 4, p. 14) and have clarified our reasoning by rephrasing the two sentences: “Thus, the quadratic effects are weighted twice as heavily as the interaction term in the computation of a_4 , masking congruence effects at the extreme ends of the predictors, as indicated by the interaction term. Hence by looking at the interaction term and the quadratic terms separately, we can obtain more detailed information about any congruence (cf. Wolf et al., 2019).”

9. P14: last line before table: Benevolence mattered least, but tradition was about the same as conformity.

We rephrased the sentence in line with the suggestion: “In contrast, fit in benevolence mattered least for well-being.” (now p. 15)

10. P19: You offer a number of explanations for why self-transcendence values did not show

the same consistent patterns as openness values with which they share the designation 'anxiety-free'. Self-transcendence values differ from openness values in their social vs. personal focus. This may provide another key to understanding the different results. For example, satisfying self-transcendence values involves reaching out to others in a supportive, closeness-building way. This may conflict with seeking to be distinctive. The social vs. personal focus may also be relevant for understanding the findings for the anxiety avoidance values. Security values have a mixed social and personal focus, conformity and tradition do not.

We thank the Reviewer for this suggestion and added it in the Discussion “Also, we found consistent congruence effects mainly for values with a personal focus (Schwartz et al., 2012): self-direction, stimulation, hedonism, achievement, power, and partly security. Those values are concerned with the outcome for oneself and are therefore conceptually more closely linked with indicators of personal well-being.” (p. 20f).

11. P20: The first two sentences are confusing. Do you mean values in regions were similar to those of their country? Please clarify. If you mean little variation in values across countries, how could there be meaningful differences in country level values.

We are indeed referring to little variation in values across countries. This analysis replicates the main finding of Fischer and Schwartz's (2011) who found that “country” only explains 2-12% of the variability in values. In other words, between-country variability is very small compared to within-country variability, resulting in high similarities of value priorities between countries (Hanel et al., 2019). This finding does not rule out the possibility of meaningful differences in country-level values: the mean differences in values between countries are small, but in line with theoretical predictions and are also reliably correlated with other variables such as level of democratization (Schwartz, 2006) or peacefulness of countries (Basabe & Valencia, 2007). We now clarify this point in Footnote 5 on page 21: “In other words, there is little variation in values between countries but substantial variation within countries (Fischer & Schwartz, 2011). However, these small mean differences in values between countries are in line with theoretical predictions and are also reliably correlated with other variables, such as level of democratization (Schwartz, 2006) and peacefulness of countries (Basabe & Valencia, 2007).”

12. P20: The curvilinear association of stimulation values with well-being is implied in the

original presentation of this value in Schwartz, 1992: “Stimulation values derive from the presumed organismic need for variety and stimulation in order to maintain an optimal level of activation (Berlyne, 1960; Houston & Mednick, 1963; Maddi, 1961).” ‘Optimal’ implied that the need is not for an extremely high or low level. Your novel finding supports this idea.

We are grateful for this suggestion and now added relevant text on page 22 “This outcome was previously postulated, but not tested by Schwartz (1992, p. 7): “Stimulation values derive from the presumed organismic need for variety and stimulation in order to maintain an optimal level of activation.” Our findings support this reasoning, which implies that extremely high or low levels of stimulation might be less beneficial.”

13. P21: The fact that achievement values behaved like anxiety avoiding values in this study is interesting. They are at the border of self-enhancement and openness in the circle. I defined them as seeking approval for success, but there seems to be a tendency of some respondents to understand them as more intrinsic pursuit of successful performance. This may be less the case with the ESS items than with the PVQ versions. Another possible explanation for the congruence effect for power and achievement may be that behavior in pursuit of these two values is liable to elicit negative reactions from others because it is counter-normative. But in countries where these values are widely cherished, such behavior is more normative, so negative reactions are weaker.

We agree that this would be an interesting hypothesis to test, but it is not possible with our data, because we would otherwise include the country-level averages twice into the three-way interaction. Specifically, if we were to test whether the interaction between individuals power values and the country averages for power values would be moderated by the country averages for power, it would be the quadratic term of country averages multiplied by individual power values: $\text{power_country} \times \text{power_country} \times \text{power_individual}$. We therefore hope that our explanation for why we predicted a congruence effect for self-enhancement is sufficient: “For example, if an individual’s society values achievement (e.g., success), and the individual does as well, then the individual might feel supported by the sense of common purpose. If the individual does not value achievement, the individual might feel pressured by social norms to achieve more. This pressure, in turn, may reduce the individual’s subjective well-being.” (p. 4).

14. Why are all the simple linear effects of values positive? This is contrary to literature summarized in Schwartz & Sortheix (2017) that shows fairly consistent negative effects of tradition, conformity, security and power? This deserves some comment.

We believe that the main difference between the approach chosen by other studies and our approach is that we did not ipsatise the values on an individual level for reasons outlined in our responses to Reviewer 3 below (points 1 and 4).¹ Ipsatising can pull the strength of an association between values and external variables such as well-being towards -1. The rank order (from most positive to least positive) of the associations is comparable across previous research and our findings: In both studies, the link between openness values and well-being is strongest, whereas conservation values correlated least strongly with well-being (negative in previous research and weakly positively or uncorrelated in our findings). Unfortunately, previous research did not report the results without the covariate. Thus, we cannot compare the linear-effects of our study with previous research. The correlations between the centred value scores and the well-being variables was then in line with the literature.

15. It would be helpful to add to the supplement some explanation of the multilevel analysis used and more on the polynomial method, so readers do not have to search sources if they wish to understand in more depth.

We have now provided a step-by-step description of our analytical approach in the supplemental materials.

“We provide here a step-by-step description of our analytical approach.

- 1) We T-transformed the six dependent variables (standard scores with $M = 50$ and $SD = 10$), because T-scores can be used as an effect size where a 2-point difference represents a small effect, a 5-point difference a medium effect, and an 8-point difference a large effect . Note that this does not change the pattern of correlation ($r_s = 1$ between untransformed and transformed variables).

¹ The terms “centring” and “ipsatising” are sometimes used interchangeable in the literature, which can result in confusion as it is possible to centre (ipsatise) across a variable or across the responses of a participant. We use in this cover letter to “centering” when we refer to subtracting the mean of a variable from each of its numerical value; for example, security – mean(security), because centering is commonly used in this context in the literature on, for example, moderated regressions (e.g., Aiken & West, 1991). In contrast, we use “ipsatising” when referring to ‘centering’ within participants. For a description of the latter see Schwartz (2003).

- 2) We mean-centered all 10 value types in line with guidelines for moderated regression analyses (Aiken & West, 1991). For example, we computed the mean of security and subtracted it from the security score of each participant.
- 3) We computed the country average for each of the ten value types, using the mean-centred value scores.
- 4) We performed 10 (value types) \times 6 (well-being variables) \times 2 (country vs regions) = 120 multi-level polynomial regression. Specifically, we entered the linear terms of individual value scores, the country (region) averages, the two respective quadratic terms which we obtained by multiplying separately each of the two linear terms with itself, and the interaction term as predictors. Additionally, we included the sample size of the country (region) as covariate and country (region) as random intercept. However, note that the pattern of result remained similar when we removed the control variable and the random effect (i.e., when we performed a standard polynomial regression).”

We hope that this description together with the discussion of Figure 2 in the Introduction and the Data Analysis subsection is sufficient for the reader to get a better understanding of our analytical approach.

Reviewer 3

This manuscript examines whether objective value similarity at the regional and country-level relate to individuals' well-being. The paper draws on the European Social Survey vast dataset, including over 50,000 individuals, and uses a methodology that has not been applied to analyse this issue: multilevel polynomial regression and surface response analysis.

The main findings are that people who highly value self-direction, stimulation, and hedonism, and who live in countries and regions where these values are also highly valued on average report lower well-being than people who ascribe lower importance to these values and live in countries and regions where people on average give low importance to these values. In this case dissimilarity increases WB. We found the reversed pattern for achievement, power, and

security values. That is individuals who value these values highly are more satisfied when they live in regions/countries where these values are also highly valued.

In summary, the main problems I find are with the centering procedure for values and lack of country-level controls for socio-economic development. Also, it is unclear the centering used for testing the cross-level interactions and how this would influence the interpretations.

Furthermore, I'm missing the mechanisms that would explain your findings.

The reviewer's points above are elaborated in the reviewers' subsequent comments. We respond to each comment below.

On a minor note: The expression that they live in places where "people share their values" in the abstract is misleading because the authors acknowledge that within-country variability of values is very high. Thus, we cannot interpret that those who value OP highly necessarily interact with others who also value it highly.

We take the reviewer's point that our previous statement was an oversimplification and added "on average" to the sentence which now reads "People who value self-direction, stimulation, and hedonism more and live in countries and regions where people on average share these values reported lower well-being."

Introduction

1. The authors formulate expectations for openness to change values at the individual and country-level, which would be positive for well-being. But why are there no expectations for the conservation values? In past studies using large representative samples those have been consistently linked to lower well-being (e.g., Bobowik et al., 2010; Sørtheix & Lönnqvist, 2014, or Sørtheix & Schwartz, 2017). Also collectivism at the country level correlates with lower SWB. So, I wonder why only openness was mentioned here?

The main reason why we only mentioned openness is because we are mainly interested in congruence effects rather than linear effects, which have been investigated elsewhere, as Reviewer 3 states. More to the point, we did not make predictions about conservation values because the rationale for negative correlations is in our view not clear. Specifically, the negative correlations only appear in the literature when the responses were ipsatised (i.e., centring on an individual-level: subtracting the mean of all value scores from the individual scores). As we outline to the Reviewer's 4th point below, we believe that

ipsatising is problematic. Without ipsatising the value scores, conservation values tend to be uncorrelated – or sometimes even slightly positively correlated – with well-being (see Tables S1-S4). This finding is much more in line with abundant research on the associations between religiosity (a conservation value of Schwartz’s model) and well-being, which has found weak positive associations between religiosity and well-being (Churchill et al., 2019; Hackney & Sanders, 2003; Witter et al., 1985). Those researchers have typically not ipsatised the responses. Conservation values are positively associated with religiosity (with and without ipsatising; Saroglou et al., 2004; Schwartz & Huismans, 1995, and our own analysis, using the European Social Survey, round 6, data).

In sum, the evidence as of whether there should be negative associations between conservation values and well-being is mixed. As linear effects are not the main focus of the paper, we decided to not get sidetracked by a discussion such as the one above, but are happy to include it (e.g., in a footnote) if you or the Reviewer think it is worth it.

2. It is also not clear why they expected that only value congruence in conservative values would predict well-being? As the authors wrote one could also expect similarity to be beneficial for all values, similarly to personality traits findings (another reference: Fulmer, C. A., Gelfand, M. J., Kruglanski, A. W., Kim-Prieto, C., Diener, E., Pierro, A., & Higgins, E. T. (2010). On “feeling right” in cultural contexts: How person-culture match affects self-esteem and subjective well-being. Psychological Science, 21(11), 1563-1569.]

The revised paper elaborates on the option that all value types might predict well-being on page 8f: “It is conceivable that well-being is supported by person-country value congruence, regardless of the type of value being examined, because the environmental affordances and social sanctions operate similarly for all values. For example, people whose values deviate from the prevalent values in a society might be socially sanctioned, regardless of value type. Alternatively, Schwartz’s (1992; Schwartz et al., 2012) model predicts that values’ associations with other variables follow a sinusoidal pattern across values, such that values that serve opposing motivations in his circular model will exhibit an opposite pattern of correlations with other variables. From this perspective, value congruence at opposing ends of the value circle may exhibit opposing patterns of relations with well-being.”

We therefore predicted partly different congruence effects for openness, self-enhancement, and conservation values. As we outlined in the Introduction, we assumed a positive interaction for conservation values and a negative interaction for openness values.

However, we did not see a theoretical reason to assume congruence effects for self-transcendence values, so we kept it exploratory.

Importantly, we found that the sign of the interaction is moderated by value type. This goes beyond research on personality traits, which only found interactions in one direction (or no significant interaction) (2016). This shows that the content of Schwartz's value model is different to the content of the Big-5 (some personality researchers seem to neglect values because they appear to consider them very similar to values; e.g., Bilsky & Schwartz, 1994).

Methods

3. The authors have excellent methodological knowledge and cite all appropriate references. However, some comments. In the description of the value measure, the PVQ21, the authors do not tell if they reversed the items so that higher scores would imply higher endorsement of the value (I assume they did this).

We thank the reviewer for spotting this oversight. We now clarified on page 12 "Prior to performing any analysis, we recoded all 21-items so that higher scores indicate greater endorsement."

4. More importantly, the authors don't tell whether they corrected for scale use by centering participants' responses around their own mean, which converts values into priorities. I think it is a serious problem if the authors did not do this. I assume they didn't, as all values have positive coefficients with the outcomes instead of following a pattern of negative to positive correlation as the expected sinusoid curve would anticipate.

The reviewer is correct in assuming that we did not ipsatise the value scores (see our response to Reviewer 2 for the distinction between centring and ipsatising).

We base this decision on several recent observations. First, He and van de Vijver (2015) argue against the reviewer's claim that ipsatising ("centering") corrects for scale use, concluding from the results of their studies "[s]core corrections to deal with response styles are not recommended" (p. 129).

Second, He et al. (2017) found that ipsatisation reduces internal consistency and measurement invariance, which is especially important for cross-cultural research (note that Davidov et al., 2008, do not report to have ipsatised the value scores prior to testing for measurement invariance across countries).

Third, we already centred the data on a variable level as recommended by guidelines for moderated regression analysis (Aiken & West, 1991). Ipsatising and then centring the data would reduce the interpretability of the data.

Fourth, Borg and Bardi (2016) reported that the mean rating of the value responses of each person is correlated with well-being. That is, how much participants agreed on average with *all* value items correlated positively with well-being: People who are depressed are less likely to find any value important. Replicating the findings of Borg and Bardi in the dataset we are using yielded in correlations between the mean ratings and the six well-being dimensions of $r = .04$ (communal well-being) and $r = .33$ (functioning). Thus, if we were to subtract from the value scores the mean rating, which is itself is correlated with well-being, findings would be distorted.

As a smaller consideration, the sinusoidal curve mentioned by the reviewer emerges to a very similar extent for the raw and ipsatised value scores (Hanel et al., 2017). The main difference is that the curve can be shifted up- or downwards along the y-axis, depending on whether raw or ipsatised scores are used. Thus, we hope you agree that it is better not to include ipsatised scores in this paper.

5. In the results, this could explain that security is “positively” related to well-being (page 15) even though it has the smallest positive coefficient of all values (Annex) and if values would be centered as in most research using the Schwartz, I assume security would be negative. This is even more relevant if we want to compare these results with past studies using the same ESS dataset and showing security was negatively related to WB (e.g., Bobowik et al., 2010; Sortheix & Lönnqvist, 2014, or Sortheix & Schwartz, 2017). Those studies used centered value scores and results (although not perfectly) follow the expected sinusoid curve: security, power were negatively associated with well-being while hedonism and benevolence were positively related, in general (life satisfaction, negative, positive emotions).

Yes, the ipsatised security score is negatively correlated with some well-being measures. Please see our response to Reviewer 3’s points 1 and 4 as well as Reviewer 2’s point 14 for explanations why we did not ipsatise for the polynomial analysis of well-being. In short, ipsatising decreases internal consistency, hampers interpretability of the outcomes, and the findings regarding the associations between the ipsatised conservation value scores and well-being contradict findings that found positive associations between religiosity and well-being.

6. Ideally, the cross-cultural equivalence for scales for well-being should have been assessed, but I think that with such a large number of items this is difficult. It should at least be mentioned in limitations. Particularly, I wonder what this scale with 14 items really measures and if there is any validity for it: “The 14-item functioning scale asked whether participants were free to decide how to live their life, were enthusiastic about what they are doing, or felt like a failure (recoded; $\alpha = .85$). I know of one study that found very limited validity by the measurement proposed by Jeffrey et al. (2015) that you used. [Charalampi, A., Michalopoulou, C., & Richardson, C. (2018). Validation of the 2012 European Social Survey Measurement of Wellbeing in Seventeen European Countries. *Applied Research in Quality of Life*, 1-33.]

We thank the Reviewer for pointing us to the paper by Charalampi et al. (2020; the online advance version was published in 2018). Charalampi et al. tested the factor structure using the six-dimensional scale from Jeffrey et al. (2015), which we also used, in 17 out of the 29 countries (using the same dataset, the ESS6). They found evidence for a four-factor solution in 7 countries, evidence for five-factors in 8 countries, and evidence for six-factors in 2 countries. Charalampi et al. concluded that “the analysis did produce reliable and valid summary measures (subscales) of wellbeing for informing social policy in each country” (p. 73). Across all countries, the six well-being scales correlate on average highly with each other (median correlation $r = .52$).

We agree with the reviewer that assessing equivalence for a large number of items across many dimensions is difficult. We followed the Reviewer’s suggestion and added this as a limitation on page 22: “One limitation of our study pertains to the six-dimensional well-being measure (Jeffrey et al., 2015), which is likely not invariant across countries: Charalampi et al. (2020) analysed the six-dimensional well-being measure in 17 out of the 29 countries that participated in the 6th round of the European Social Survey. They found the best statistical fit for a four-factor solution in some countries, a five-factor solution in other countries, and a six-factor solution in only two countries. However, we do not expect different results if the items were combined into different factors, because the six dimensions correlated highly with each other, on average (median $r = .52$). Further, it is typically difficult to establish measurement invariance across a larger set of items, factors, and especially countries. Indeed, Charalampi et al. concluded that “the analysis did produce reliable and

valid summary measures (subscales) of wellbeing for informing social policy in each country” (p. 73).”

7. *On page 14, please, open up for the reader what you mean by: “Following Bleidorn et al. (2016), we did not test formally for congruence (Humberg et al., 2018)”.*

When writing the previous version of the manuscript, we were unsure whether we should keep this comment or remove it because formally testing for congruence is complex. We decided to keep this remark in the submitted manuscript for transparency. According to Humberg et al., there are “four conditions to test congruence effects in a broad sense and six conditions to test congruence effects in a strict sense” (p. 3). Discussing these conditions would go well beyond our paper. To clarify, we now added to the manuscript “Following Bleidorn et al. (2016), we did not formally test for congruence. Specifically, Edwards (2002; see also Humberg et al., 2018) argued that, among other things, a strict requirement for congruence is that the ridge of the plane needs to be exactly above the main diagonal of the plot, that is between the points (-2|-2|-4) and points (2|2|-4), as it is the case for both planes displayed in Figure 2. This requirement is difficult to achieve in the present research because the variance for the country-level scores is significantly smaller than for the individual-level value scores.” (p. 14f)

8. *The centering procedure for all variables is not sufficiently explained. Did you grand or group-mean centered?*

As noted in response to Reviewer 2, Point 15, we centred predictors along their respective mean. Following Reviewer 2’s request, we now provide a more detailed description of our analysis in the Supplemental Materials.

Results

9. *Page 15: The relation between security and evaluative well-being is absent in countries in which people value security less on average than in countries in which security is more valued, $B_s = 0.00$ vs 0.43 . “ Does the interpretation also imply that when living in countries where security is less valued they report lower well-being? If we think about this in real life, this would mean that a person who values highly security (which as I wrote before predicts lower WB), and lives in Poland or Italy (high on security in ESS), he/she is more satisfied*

than if she/he lives in Finland (relatively low on security)? Those countries high on security are generally characterized by their instability, making life more insecure (contradicting the goals of conservative values), as we know that more socio-economic development correlates with higher individualism. That argument was used by past research to argue why the relation of security and other conservation values was more strongly negative in countries with lower HDI (Sortheix & Lönnqvist, 2014) or less egalitarian (Sortheix & Schwartz, 2017). However, I fail to see what mechanism would explain your findings. Have you tested if your results are robust when you control for socio-economic development?

This comment consists of two main parts. First, “Does the interpretation also imply that when living in countries where security is less valued they report lower well-being” refers to a negative linear association between country-level values and well-being. Indeed, we report evidence for this negative linear association in the Supplemental Materials (although some of the linear terms did not reach statistical significance at $\alpha = .001$). Note, however, that individual-level security scores were inconsistently associated with the six well-being variables (e.g., Table S1). This country-level effect is relatively strong. As can be seen in Figure 3A, well-being tends to be lowest in countries in which people value on average security more (in line with the Reviewer’s predictions). However, there is a little upwards bend at the end of the plain (towards point 2|2|66), which qualifies the interaction. That is, people’s well-being is slightly higher if they score high on security and live in countries in which people score high on security as well, than one would expect if there were only linear and quadratic effects. To clarify, we write in the Results section “The similarity effect can be mainly seen at the $x=2|y=2$ point where well-being is higher than could be expected based on the two linear term effects. This interpretation is further supported by a simple slope analysis.” (p. 16f)

We thank the Reviewer for suggesting that we include socio-economic development as a control variable. We now added on page 13f: “Also, controlling for the country-level variable Human Developmental Index (United Nations Developmental Programme, 2014), individual-level education, or income resulted in the same pattern of findings (Table S5).” And in a footnote to this we added “Specifically, including HDI as a country-level control variable left the individual-level effects and, more importantly, interaction terms unaffected (see Table S5). It reduced, however, the country-level effects substantially. These are less relevant for the presented research (the reduced country-level effects can be explained by

previous research suggesting strong associations between country-level scores of well-being and other country-level variables such as income; Diener et al., 1995). The finding that HDI does not affect the congruence effects is echoed by our supplemental analysis where we found that the pattern of results was very similar across Eastern and Western European countries (see Online Supplemental Materials).

However, given that there is substantial variation in terms of education level and income within each country, using country-level estimates might be less appropriate. We therefore tested the robustness of our effects using individual-level education level using the International Standard Classification of Education and the income. Income was measured in a country-specific way, by coding the total household's net income into the respective decile of the income distribution for the household's nation. Both variables were reported in the dataset we used. However, adding those two variables as controls again had little impact on our findings." (Footnote 2, p. 14)

10. We observed a similar pattern for power predicting supportive relationships" And for the other outcomes? The interaction was significant for all outcomes. So, I'm not sure why you mention only this one.

We performed 120 multi-level polynomial regression analyses to test our main hypotheses (not counting the analyses with the different control variables). We selectively displayed four of them in Figures 3A-3D. We selected two value types for which a negative interaction term emerged (self-direction and stimulation) and two value types for which we found a positive interaction (power and security), because the pattern of results was consistent for those four value types across all six dependent variables. We discuss the findings in detail for stimulation and self-direction in the Result section. If you or the Reviewer feel that adding more information about other value types or dependent variables, including graphs, would be useful, we are happy to comply.

11. How do you explain your findings for openness? Your results would mean that a person who highly values openness and lives in Denmark (high in OP) or Copenhagen region (high in OP) would report lower well-being.... This doesn't make sense to me. The interpretation in the discussion also goes against past research showing that people enjoy higher well-being in anxiety-free contexts, e.g., more autonomous, self-expression societies (see Schwartz, Inglehart and Welzel model of human development, Fischer and Boer's findings). So, I find

this sentence ungrounded: “Our findings indicate that adaptation to prevalent anxiety-free values is linked to lower well-being, and adaptation to prevalent avoidance-focused values is linked to higher well-being. “ It would be then important to show whether results hold after including HDI or GDP and offer a more convincing explanation.

The interaction for the openness values indicate that, to put it simple, the plane does not follow a linear trend upwards, but has a bend downwards towards its end (i.e., towards $x = 2$, $y = 2$). That is, people valuing openness values and living in countries or regions in which, on average, openness values are relatively high, report higher well-being. This is evidenced by strong individual-level and country-level linear effects (they are stronger for openness values than for the other values). However, the negative interaction reveals that the association between openness values and well-being is stronger when openness values are below average (vs. above average). As we reported above, our findings are robust when controlling for HDI or individual-level income and education (GDP is included in the HDI).

12. In my view, for this paper to have more influence in the field, it would be still needed /useful to compare and offer explanations that enlighten other studies using the same ESS dataset which also found dissimilarity effects using other indicators at the country-level. For instance, Sortheix and Lönnqvist (2014) used HDI and found that individuals high in OP values who live in low HDI countries (which are generally lower on OP at the country-level) report higher life satisfaction. It would be necessary to find a novel argument for what you found.

Previous research (e.g., Sortheix and Lönnqvist, 2014) tested whether HDI moderates the associations between values and well-being. In contrast, we tested the person-environment fit hypothesis, which is conceptually and methodologically different. First, whether one’s values align or don’t align with other people’s value is independent of HDI or other indices. Also, any of the values are conceptually different to years of schooling, life expectancy, and income (i.e., the HDI). Thus, an interaction between individual-level values and country-level values is quite different to an interaction with HDI or any other indices. As we have shown above, controlling for HDI did not affect the pattern of results.

It is also methodologically different, as a polynomial regression and response surface analysis are not the same as a moderated regression analysis. The main difference is that, to test effects of value congruence or value fit, one needs to control for the curvilinear terms, to

ensure that the interaction “does not spuriously reflect curvilinearity”, which is a common issue in moderated regression (J.R. Edwards, 2008, p. 152).

Additional change

We also made a small change to our analytical strategy, which had, however, very little impact on the outcome: In the previous version of the manuscript we controlled, following Bleidorn et al. (2016), individual-level age and gender, and on the country- and regions-level, we additionally controlled for mean age and sample size. However, we reconsidered our analytical strategy and excluded age and gender as control variables and only kept country- or region-level sample size because we wanted to adjust for differences in sample sizes. More specifically, the reviewer feedback led us to question our own rationale “we need to control for age and gender because they are correlated with values or well-being”. This raises the question: Why control for them, but not for all the other variables that are also and potentially more strongly associated with values and well-being? Especially in light of representative samples such as the one we are using, controlling for age and gender appears unnecessary. Further, many studies use different control variables, hindering the comparability of studies. We therefore decided to drop controls as far as possible. However, we still report the results of the polynomial regressions with those control variables in Tables S3 and S4 and write in the Data Analysis subsection “We controlled for country- or regions-level sample size, following Bleidorn et al. (2016). The pattern of results, however, remained similar when we controlled for individual-level age and gender as well as country-level age: correlations between individual-level values without and with controlling for age and gender were $r(64) = .97$, and correlations between the interaction terms were $r(64) = .99$ (see tables S3 and S4).” Overall, we are impressed that the patterns we report are robust across a number of options for statistical control and analysis (as shown in the Supplements), which may attest to both the strength of the psychological processes we are revealing and to the utility of the polynomial approach for capturing congruence effects.

To conclude, we are grateful for the constructive feedback from you and the reviewers. We have carefully considered each and every point and believe that addressing this feedback has improved the paper significantly. We hope you will now find it acceptable for publication in *Nature Communication*.

With kind regards

Paul Hanel, Uwe Wolfradt, Lukas Wolf, Gabriel Coelho, Greg Maio

References

- Aiken, L. S., & West, S. G. (1991). *Multiple regression: Testing and interpreting interactions*. Sage Publications.
- Basabe, N., & Valencia, J. (2007). Culture of peace: Sociostructural dimensions, cultural values, and emotional climate. *Journal of Social Issues, 63*(2), 405–419.
<https://doi.org/10.1111/j.1540-4560.2007.00516.x>
- Bilsky, W., & Schwartz, S. H. (1994). Values and personality. *European Journal of Personality, 8*(3), 163–181. <https://doi.org/10.1002/per.2410080303>
- Bleidorn, W., Schönbrodt, F., Gebauer, J. E., Rentfrow, P. J., Potter, J., & Gosling, S. D. (2016). To live among like-minded others: Exploring the links between person-city personality fit and self-esteem. *Psychological Science, 27*(3), 419–427.
<https://doi.org/10.1177/0956797615627133>
- Boehnke, K. (2001). Parent-offspring value transmission in a societal context: Suggestions for a utopian research design—with empirical underpinnings. *Journal of Cross-Cultural Psychology, 32*(2), 241–255. <https://doi.org/10.1177/0022022101032002010>
- Borg, I., & Bardi, A. (2016). Should ratings of the importance of personal values be centered? *Journal of Research in Personality, 63*, 95–101.
<https://doi.org/10.1016/j.jrp.2016.05.011>
- Charalampi, A., Michalopoulou, C., & Richardson, C. (2020). Validation of the 2012 European Social Survey measurement of wellbeing in seventeen european countries. *Applied Research in Quality of Life, 15*(1), 73–105. <https://doi.org/10.1007/s11482-018-9666-4>
- Churchill, S. A., Appau, S., & Farrell, L. (2019). Religiosity, income and wellbeing in developing countries. *Empirical Economics, 56*(3), 959–985.
<https://doi.org/10.1007/s00181-017-1380-9>

- Davidov, E., Schmidt, P., & Schwartz, S. H. (2008). Bringing values back in: The adequacy of the European Social Survey to measure values in 20 countries. *The Public Opinion Quarterly*, *72*(3), 420–445.
- Diener, E., Diener, M., & Diener, C. (1995). Factors predicting the subjective well-being of nations. *Journal of Personality and Social Psychology*, *69*(5), 851–864.
- Edwards, Jeffrey R. (2002). Alternatives to difference scores: Polynomial regression analysis and response surface methodology. In F. Drasgow & N. W. Schmitt (Eds.), *Advances in measurement and data analysis* (pp. 350–400). Jossey-Bass.
- Edwards, Jeffrey R., & Cable, D. M. (2009). The value of value congruence. *Journal of Applied Psychology*, *94*(3), 654–677. <https://doi.org/10.1037/a0014891>
- Edwards, J.R. (2008). Seven deadly myths of testing moderation in organizational research. In *Statistical and Methodological Myths and Urban Legends: Doctrine, Verity and Fable in the Organizational and Social Sciences* (pp. 143–164). Scopus.
<https://doi.org/10.4324/9780203867266>
- Eriksson, E. L., Becker, M., & Vignoles, V. L. (2011). Just another face in the crowd? Distinctiveness seeking in Sweden and Britain. *Psychological Studies*, *56*(1), 125–134.
<https://doi.org/10.1007/s12646-010-0030-5>
- Fischer, R., & Schwartz, S. (2011). Whence differences in value priorities? Individual, cultural, or artifactual sources. *Journal of Cross-Cultural Psychology*, *42*(7), 1127–1144. <https://doi.org/10.1177/00220221110381429>
- Fulmer, C. A., Gelfand, M. J., Kruglanski, A. W., Kim-Prieto, C., Diener, E., Pierro, A., & Higgins, E. T. (2010). On “Feeling Right” in cultural contexts: How person-culture match affects self-esteem and subjective well-being. *Psychological Science*, *21*(11), 1563–1569. <https://doi.org/10.1177/0956797610384742>

- Hackney, C. H., & Sanders, G. S. (2003). Religiosity and Mental Health: A Meta-Analysis of Recent Studies. *Journal for the Scientific Study of Religion*, 42(1), 43–55.
<https://doi.org/10.1111/1468-5906.t01-1-00160>
- Hanel, P. H. P., Maio, G. R., & Manstead, A. S. R. (2019). A new way to look at the data: Similarities between groups of people are large and important. *Journal of Personality and Social Psychology*, 116(4), 541–562. Scopus.
<http://dx.doi.org/10.1037/pspi0000154>
- Hanel, P. H. P., & Vione, K. C. (2016). Do student samples provide an accurate estimate of the general public? *PLOS ONE*, 11(12), e0168354.
<https://doi.org/10.1371/journal.pone.0168354>
- Hanel, P. H. P., Wolfradt, U., Coelho, G. L. de H., Wolf, L. J., Vilar, R., Monteiro, R. P., Gouveia, V. V., Crompton, T., & Maio, G. R. (2018). The perception of family, city, and country values is often biased. *Journal of Cross-Cultural Psychology*, 49(5), 831–850. <https://doi.org/10.1177/0022022118767574>
- Hanel, P. H. P., Zacharopoulos, G., Mégardon, G., & Maio, G. R. (2017). Detecting sinusoidal patterns from circumplex models of psychological constructs. *Preprint*.
<https://psyarxiv.com/wh92k/>
- He, J., & van de Vijver, F. J. R. (2015). Self-presentation styles in self-reports: Linking the general factors of response styles, personality traits, and values in a longitudinal study. *Personality and Individual Differences*, 81, 129–134.
<https://doi.org/10.1016/j.paid.2014.09.009>
- He, J., Van de Vijver, F. J. R., Fetvadjev, V. H., de Carmen Dominguez Espinosa, A., Adams, B., Alonso-Arbiol, I., Aydinli-Karakulak, A., Buzea, C., Dimitrova, R., Fortin, A., Hapunda, G., Ma, S., Sargautyte, R., Sim, S., Schachner, M. K., Suryani, A., Zeinoun, P., & Zhang, R. (2017). On enhancing the cross-cultural comparability of

- Likert-scale personality and value measures: A comparison of common procedures. *European Journal of Personality*, 31(6), 642–657. <https://doi.org/10.1002/per.2132>
- Higgins, E. T. (2019). *Shared reality: What makes us strong and tears us apart*. Oxford University Press.
- Humberg, S., Nestler, S., & Back, M. D. (2018). Response surface analysis in personality and social psychology: Checklist and clarifications for the case of congruence hypotheses. *Social Psychological and Personality Science*, 1948550618757600. <https://doi.org/10.1177/1948550618757600>
- Jeffrey, K., Abdallah, S., & Quick, A. (2015). *Europeans' personal and social wellbeing: Topline results from Round 6 of the European Social Survey*. https://www.europeansocialsurvey.org/docs/findings/ESS6_toplines_issue_5_personal_and_social_wellbeing.pdf
- Montoya, R. M., Horton, R. S., & Kirchner, J. (2008). Is actual similarity necessary for attraction? A meta-analysis of actual and perceived similarity. *Journal of Social and Personal Relationships*, 25(6), 889–922. <https://doi.org/10.1177/0265407508096700>
- Peterson, R. A. (2001). On the Use of College Students in Social Science Research: Insights from a Second-Order Meta-analysis. *Journal of Consumer Research*, 28(3), 450–461.
- Roccas, S., Schwartz, S. H., & Amit, A. (2010). Personal value priorities and national identification. *Political Psychology*, 31(3), 393–419. <https://doi.org/10.1111/j.1467-9221.2010.00763.x>
- Roest, A. M. C., Dubas, J. S., Gerris, J. R. M., & Engels, R. C. M. E. (2009). Value Similarities Among Fathers, Mothers, and Adolescents and the Role of a Cultural Stereotype: Different Measurement Strategies Reconsidered. *Journal of Research on Adolescence*, 19(4), 812–833. <https://doi.org/10.1111/j.1532-7795.2009.00621.x>
- Sagiv, L., & Schwartz, S. H. (2000). Value priorities and subjective well-being: Direct relations and congruity effects. *European Journal of Social Psychology*, 30(2), 177–

198. [https://doi.org/10.1002/\(SICI\)1099-0992\(200003/04\)30:2<177::AID-EJSP982>3.0.CO;2-Z](https://doi.org/10.1002/(SICI)1099-0992(200003/04)30:2<177::AID-EJSP982>3.0.CO;2-Z)
- Saroglou, V., Delpierre, V., & Dernelle, R. (2004). Values and religiosity: A meta-analysis of studies using Schwartz's model. *Personality and Individual Differences*, 37(4), 721–734. <https://doi.org/10.1016/j.paid.2003.10.005>
- Schwartz, S. H. (1992). Universals in the content and structure of values: Theoretical advances and empirical tests in 20 countries. *Advances in Experimental Social Psychology*, 25, 1–65. [https://doi.org/10.1016/S0065-2601\(08\)60281-6](https://doi.org/10.1016/S0065-2601(08)60281-6)
- Schwartz, S. H. (2003). *Instructions for computing scores for the 10 human values and using them in analyses*.
http://www.europeansocialsurvey.org/docs/methodology/ESS1_human_values_scale.pdf
- Schwartz, S. H. (2006). A theory of cultural value orientations: Explication and applications. *Comparative Sociology*, 5(2), 137–182. <https://doi.org/10.1163/156913306778667357>
- Schwartz, S. H. (2018). Schwartz, Shalom. In V. Zeigler-Hill & T. K. Shackelford (Eds.), *Encyclopedia of Personality and Individual Differences* (pp. 1–3). Springer International Publishing. https://doi.org/10.1007/978-3-319-28099-8_2327-1
- Schwartz, S. H., & Bardi, A. (2001). Value hierarchies across cultures taking a similarities perspective. *Journal of Cross-Cultural Psychology*, 32(3), 268–290.
<https://doi.org/10.1177/0022022101032003002>
- Schwartz, S. H., Cieciuch, J., Vecchione, M., Davidov, E., Fischer, R., Beierlein, C., Ramos, A., Verkasalo, M., Lönnqvist, J.-E., Demirutku, K., Dirilen-Gumus, O., & Konty, M. (2012). Refining the theory of basic individual values. *Journal of Personality and Social Psychology*, 103(4), 663–688. <https://doi.org/10.1037/a0029393>

- Schwartz, S. H., & Huismans, S. (1995). Value priorities and religiosity in four Western religions. *Social Psychology Quarterly*, 58(2), 88–107. JSTOR. <https://doi.org/10.2307/2787148>
- Solomon, S., & Knafo-Noam, A. (2007). Value similarity in adolescent friendships. In T. C. Rhoades (Ed.), *Focus on adolescent behavior research* (pp. 133–155). Nova Science.
- Stromberg, C., & Boehnke, K. (2001). Person/society value congruence and well-being: The role of acculturation strategies. In P. Schmuck & K. M. Sheldon (Eds.), *Life goals and well-being* (pp. 37–57). Hogrefe.
- United Nations Developmental Programme. (2014). *Human Developmental Report: Human Development Index (HDI)*. <http://hdr.undp.org/en/data>
- Witter, R. A., Stock, W. A., Okun, M. A., & Haring, M. J. (1985). Religion and Subjective Well-Being in Adulthood: A Quantitative Synthesis. *Review of Religious Research*, 26(4), 332–342. JSTOR. <https://doi.org/10.2307/3511048>
- Yi, C.-C., Chang, C.-F., & Chang, Y.-H. (2004). The intergenerational transmission of family values: A comparison between teenagers and parents in Taiwan. *Journal of Comparative Family Studies*, 35(4), 523–545. JSTOR.

REVIEWER COMMENTS

Reviewer #1 (Remarks to the Author):

This paper now surely is in a publishable shape.

I will nevertheless wage a few comments that might motivate the authors to allow for another round of polishing.

Following is a collection of nitty-gritty and more substantial points

p. 3: Although myself coming from the 'Schwartz camp', I think it sounds a bit strong to call Shalom's model "The most prominent model of human values." There are competitors, I believe, the Maslow-Inglehart-Welzel approach to values seems similarly prominent if one looks at all social sciences. Furthermore, the late Geert Hofstede and Harry Triandis also had important words to say...

p.3: I do not think it is advisable to jump between 'circular' and 'circumplex' (p.4) to describe the model. I think you should always stick to 'circumplex' (because of the 'irregularity' of TR and CO)

p.10: "We made no prediction regarding self-enhancement and self-transcendence." My spontaneous reaction was "what a pity." Although I agree that working on this is another piece of research, I think expert readers would at least expect a brief elaboration of 'why' (either here or in the limitations and/or discussion sections).

p. 11: I was not happy with the emergence of the term 'individualism' which stems from a different set of value theory.

p. 12: I believe when the regions are introduced, it might help to use the NUTS2, NUTS3 terminology of European statisticians. How does it relate here?

p.12/13: The question arises whether these well-being measures suffice equivalence criteria. You address this issue later in the limitations section, but it should probably already be addressed here. I am not the greatest friend of overly high equivalence requirements, but dropping the issue here is a bit awkward, after you have cited Eldad's work for the PVQ.

p.15: I like your very brief statement 'if significant at .001.' This avoids the dreadful current preoccupation with 'power.' Good choice!

p.15--Table 1): Here your simple sentence on not wanting to say anything on self-transcendence and self-enhancement values fires back. Why are all ten values reported and not only the one's relevant in the context of your hypotheses. Perfectly understandable, sure, but it forces you to say something about the non-finding for BE, which probably is due to a ceiling effect (everyone cherishes benevolence

everywhere).

p.15-p.18: There is a certain imbalance between the detailedness of reporting results on individual-country fit and individual-region fit. Later in the discussion you also address supranational units (East/West). Where are the results for this. No need to report them, but in the conclusion reference to them comes out of the blue.

I like it that this is now clearly a substance paper and no longer a paper demonstrating statistical expertise (which for me was the case with a earlier version to some extent).

If I have a wish free, I would like the authors to go at least one step further in the discussion and at least briefly address the question how people succeed in aligning with value preferences of social units. This would open the door to sociology....

Klaus Boehnke

Reviewer #2 (Remarks to the Author):

lines 257-7 This reference is to a scale with 29 items that overlap only very partially with the scale used in this study. It is the first mention of the label PVQ and use of the portrait approach to measure values. It can legitimately be referenced for that purpose. However, it would be better to add a reference to the source that first presented the PVQ used in this study and that lists the items. That reference is: Schwartz, S. H. (2003). Value orientations. European Social Survey Core Questionnaire Development, Chapter 07 . Website: http://www.europeansocialsurvey.org/docs/methodology/core_ess_questionnaire/ESS_core_questionnaire_human_values.pdf

Although not critical for this paper, I would like to comment on the issue of ipsatizing value scores. I think this issue is more complex than your comments in the paper suggest.

Appropriateness of ipsatizing

Whether ipsatizing is desirable or not is as much a conceptual as a methodological issue. Critics of ipsatizing usually base their argument on its distorting effect on associations of the individual ipsatized scores and their reduced internal reliability. (BTW, reliability/consistency over time is not affected.) I agree that value scores should not be ipsatized when computing internal reliability. However, when relating them to other variables, the appropriateness of ipsatizing depends on one's aims and assumptions. I ipsatize despite He & van de Vijver/Borg & Bardi because my assumption about values leads me to emphasize a different aim/goal.

I assume that values do not affect and are not affected by most other variables separately for each value. If that were the case, it would justify the objection to ipsatizing. Rather values relate to other

variables based both through facilitating and inhibiting processes. At a minimum, relations depend on the balance between two opposing values. More often, multiple opposing values may be involved. It is the trade-off between opposing values that is relevant, a trade-off that requires considering the RELATIVE importance of the values involved for each individual. Ipsatizing converts the absolute importance of the value scores into relative importance scores.

Findings for security values are illustrative of the meaningfulness of ipsatizing value scores. The absolute importance of security to the individual, as reflected in a scale response, may not relate to his/her well-being. If security values are more, rather than less, important to that person than openness values, however, as revealed by ipsatizing, the trade-off during experiences that affect WB is likely to lower WB. That is what the literature that reports negative associations between ipsatized security scores and SWB finds.

Shalom H. Schwartz

Reviewer 1

This paper now surely is in a publishable shape.

I will nevertheless wage a few comments that might motivate the authors to allow for another round of polishing.

Following is a collection of nitty-gritty and more substantial points

p. 3: Although myself coming from the 'Schwartz camp', I think it sounds a bit strong to call Shalom's model "The most prominent model of human values." There are competitors, I believe, the Maslow-Inglehart-Welzel approach to values seems similarly prominent if one looks at all social sciences. Furthermore, the late Geert Hofstede and Harry Triandis also had important words to say...

We now write "One of the most prominent models..."

p.3: I do not think it is advisable to jump between 'circular' and 'circumplex' (p.4) to describe the model. I think you should always stick to 'circumplex' (because of the 'irregularity' of TR and CO)

We now write consistently "circumplex".

p.10: "We made no prediction regarding self-enhancement and self-transcendence." My spontaneous reaction was "what a pity." Although I agree that working on this is another piece of research, I think expert readers would at least expect a brief elaboration of 'why' (either here or in the limitations and/or discussion sections).

We now provide the requested elaboration: "We made no prediction regarding self-enhancement and self-transcendence values due to the complex and diverging literature related to agentic and communal traits, social comparison, and self-esteem. For example, on the one hand, one might assume that people high in competitiveness would thrive in a competitive environment, while those high in cooperativeness would thrive in a cooperative environment. These environments would be functional for their goals and the self-enhancement and self-transcendence values that activate them. On the other hand, evidence suggests that people socially compare on both agentic and communal traits (Gebauer et al., 2012), which may suggest lower well-being when people seem too similar in relatively agentic and communal values, particularly if these are important dimensions of comparison for social identity (Roccas & Schwartz, 1993)." (p. 10f)

p. 11: I was not happy with the emergence of the term 'individualism' which stems from a different set of value theory.

For context, the sentence in question reads “Further, on a country-level, individualism, which entails self-direction and stimulation, is positively associated with well-being (Diener et al., 1995).” The term individualism was used by Diener et al. and indeed stems from another theory (Hofstede’s cultural dimension model). To link it to Schwartz’s model, we wrote “...which entails self-direction and stimulation...”.

p. 12: I believe when the regions are introduced, it might help to use the NUTS2, NUTS3 terminology of European statisticians. How does it relate here?

NUTS is a geographical nomenclature used to subdivide EU-countries into regions. However, as governments use different NUTS levels and the dataset we are using contains several countries which are not in the EU (e.g., Israel, Kosovo), we decided not to discuss NUTS, but simply refer to region as specified by the European Social Survey, which is in line with official governmental use (e.g., 9 regions in England, 16 federal states in Germany...).

See here for more information

<https://www.europeansocialsurvey.org/data/multilevel/guide/essreg.html> The names of the regions can easily be accessed in the dataset found here

https://www.europeansocialsurvey.org/download.html?file=ESS6e02_4&y=2012 (link is also provided in the manuscript)

p.12/13: The question arises whether these well-being measures suffice equivalence criteria. You address this issue later in the limitations section, but it should probably already be addressed here. I am not the greatest friend of overly high equivalence requirements, but dropping the issue here is a bit awkward, after you have cited Eldad's work for the PVQ.

As suggested, we now moved the discussion on equivalence/invariance of the well-being measure from the Limitations (Discussion) section to the materials section and adapted it slightly: “However, the measure is likely not fully invariant across countries: Charalampi et al. (2020) analysed the six-dimensional well-being measure in 17 out of the 29 countries that participated in the 6th round of the European Social Survey. They found the best statistical fit for a four-factor solution in some countries, a five-factor solution in other countries, and a six-factor solution in only two countries. However, we do not expect different results if the items were combined into different factors, because the six dimensions correlated highly with each other, on average (median $r = .52$). Further, it is typically difficult to establish

measurement invariance across a larger set of items, factors, and especially countries. Indeed, Charalampi et al. concluded that “the analysis did produce reliable and valid summary measures (subscales) of wellbeing for informing social policy in each country” (p. 73).” (p. 13).

p.15: I like your very brief statement 'if significant at .001.' This avoids the dreadful current preoccupation with 'power.' Good choice!

Thank you.

p.15--Table 1): Here your simple sentence on not wanting to say anything on self-transcendence and self-enhancement values fires back. Why are all ten values reported and not only the one's relevant in the context of your hypotheses. Perfectly understandable, sure, but it forces you to say something about the non-finding for BE, which probably is due to a ceiling effect (everyone cherishes benevolence everywhere).

We also reported the results for self-transcendence and self-enhancement values because we believe it is important to engage with exploratory research. We hope that briefly outlining why we did not make any predictions for those values (see above) satisfies this concern. Further, we find some unexpected yet consistent effects of self-enhancement values which we believe are worth reporting. We also discuss them in the Discussion section.

p.15-p.18: There is a certain imbalance between the detailedness of reporting results on individual-country fit and individual-region fit. Later in the discussion you also address supranational units (East/West). Where are the results for this. No need to report them, but in the conclusion reference to them comes out of the blue.

As reported, the results for individual-country fit and individual-region fit are very similar. We focus in the manuscript on the individual-country fit because cross-cultural research typically uses countries as supra-level rather than region. Reporting the results of individual-region fit to the same extent as individual-country fit seems therefore redundant. However, we report the results for individual-region fit in the Supplemental Materials as well in detail (Tables S2 and S4).

We already mention the comparison between Eastern and Western European countries in the Results section: “We also found that the pattern of results was very similar between Eastern and Western European countries (see Online Supplemental Materials).” (p. 18) The

method and results for this supplemental analysis are briefly summarized in the Online Supplemental Materials.

I like it that this is now clearly a substance paper and no longer a paper demonstrating statistical expertise (which for me was the case with a earlier version to some extent).

Thank you.

If I have a wish free, I would like the authors to go at least one step further in the discussion and at least briefly address the question how people succeed in aligning with value preferences of social units. This would open the door to sociology....

Addressing this wish could be a paper on its own, because our results suggest that whether aligning one's values with the value preferences of one's social unit is beneficial for one's well-being depends on the value type (e.g., security, stimulation) and by the value preferences one's social unit place on it. Thus, a full discussion would need to be conditional on the country people in which people live, which would further complicate things.

Nevertheless, we recognise this issue is fundamentally interesting and important. Consequently, in the space available, our revised Discussion includes brief allusion to mechanisms that can mediate value alignment. We fear that any brief treatment of this issue must be partial and incomplete, but we hope that it helps signpost the reader to this exciting topic. We now write "In fact, identification is among a number of variables that may help to understand how value congruence emerges, as people might be more motivated to identify value congruence in groups with which they identify strongly (Roccas & Brewer, 2002). Value congruence may also emerge through the same broad psychological processes that generate person-situation similarity through situation perception, selection, and adaptation in the service of chronic and acute motivations (Snyder & Cantor, 1998)." (p. 22)

More importantly, even while it might be beneficial for one's well-being to increase, for example, one's security values if people in one's country would value security a lot, there might be costs to it: Valuing security is associated with more negative attitudes towards immigrants (Davidov et al., 2008). As long as it is unclear whether increasing security values leaves attitudes towards immigrants unchanged, we do not feel that we should make any recommendations for people to change their values.

Reviewer 2

lines 257-7 This reference is to a scale with 29 items that overlap only very partially with the scale used in this study. It is the first mention of the label PVQ and use of the portrait approach to measure values. It can legitimately be referenced for that purpose. However, it would be better to add a reference to the source that first presented the PVQ used in this study and that lists the items. That reference is:

*Schwartz, S. H. (2003). Value orientations. European Social Survey Core Questionnaire Development, Chapter 07 . Website:
http://www.europeansocialsurvey.org/docs/methodology/core_ess_questionnaire/ESS_core_questionnaire_human_values.pdf*

We now added the suggested reference on page 12.

Although not criticaL for this paper, I would like to comment on the issue of ipsatizing value scores. I think this issue is more complex than your comments in the paper suggest.

Appropriateness of ipsatizing

Whether ipsatizing is desirable or not is as much a conceptual as a methodological issue. Critics of ipsatizing usually base their argument on its distorting effect on associations of the individual ipsatized scores and their reduced internal reliability. (BTW, reliability/consistency over time is not affected.)

I agree that value scores should not be ipsatized when computing internal reliability. However, when relating them to other variables, the appropriateness of ipsatizing depends on one's aims and assumptions. I ipsatize despite He & van de Vijver/Borg & Bardi because my assumption about values leads me to emphasize a different aim/goal.

I assume that values do not affect and are not affected by most other variables separately for each value. If that were the case, it would justify the objection to ipsatizing. Rather values relate to other variables based both through facilitating and inhibiting processes. At a minimum, relations depend on the balance between two opposing values. More often, multiple opposing values may be involved. It is the trade-off between opposing values that is relevant, a trade-off that requires considering the RELATIVE importance of the values involved for each individual. Ipsatizing converts the absolute importance of the value scores into relative importance scores.

Findings for security values are illustrative of the meaningfulness of ipsatizing value scores. The absolute importance of security to the individual, as reflected in a scale response, may not relate to his/her well-being. If security values are more, rather than less, important to that person than openness values, however, as revealed by ipsatizing, the trade-off during experiences that affect WB is likely to lower WB. That is what the literature that reports negative associations between ipsatized security scores and SWB finds.

We agree that ipsatising (or centering) can be interesting for some research questions. For a number of direct relations with values, it makes conceptual sense to focus on the relative importance of values rather than the absolute importance of each value or set of values. We believe that focusing on the absolute importance of values is more meaningful in the context of our study because our focus here expands from a link between values and an outcome – which is the focus of the reviewer’s example – to a more complex interrelation between individual values, other people’s values, and an outcome (well-being). The interpersonal aspect invites an empirical question about whether “values do not affect and are not affected by most other variables separately for each value”. We cannot be confident in assuming that the same social factors affect social differences the same way for different values. Furthermore, we do not yet know the role of the relatively simple social comparison, “Is security as important to other people as it is to myself?” to the relatively meta-social comparison, “Is security relative to all of other people’s values as important as it is to myself, again relative to all other values?”

On top of those conceptual arguments in favour of the absolute importance, using the relative importance instead would hinder comparisons with potential other studies that focus, for example, on effects of congruence in moral foundations, goals, or personality traits on well-being: These constructs are consistently operationalised in terms of absolute scores, not relative scores. Additionally, whether to ipsatise or not is not only a conceptual question, but also a methodological question, as outlined in our previous response: Ipsatising reduces reliabilities and cross-cultural comparability (He et al., 2017) and removes some meaningful variance, as the importance attributed to all values is positively correlated with well-being (Borg & Bardi, 2016).

Nonetheless, we take the gist of the reviewer’s point. However, our paper does not currently include any text about centring values. We have opted to maintain the paper’s brevity and focus by not adding any discussion of the pros and cons of centring because we believe the issue merits an extended discussion elsewhere. Nonetheless, if the Editor wishes, we could include a footnote alluding to the frequent utility of ipsatised or centred scores,

while noting the greater complexity of our context and its role in determining our use of absolute scores here.

References

- Borg, I., & Bardi, A. (2016). Should ratings of the importance of personal values be centered? *Journal of Research in Personality, 63*, 95–101.
<https://doi.org/10.1016/j.jrp.2016.05.011>
- Charalampi, A., Michalopoulou, C., & Richardson, C. (2020). Validation of the 2012 European Social Survey measurement of wellbeing in seventeen european countries. *Applied Research in Quality of Life, 15*(1), 73–105. <https://doi.org/10.1007/s11482-018-9666-4>
- Davidov, E., Meuleman, B., Billiet, J., & Schmidt, P. (2008). Values and support for immigration: A cross-country comparison. *European Sociological Review, 24*(5), 583–599. <https://doi.org/10.1093/esr/jcn020>
- Diener, E., Diener, M., & Diener, C. (1995). Factors predicting the subjective well-being of nations. *Journal of Personality and Social Psychology, 69*(5), 851–864.
- Gebauer, J. E., Sedikides, C., Verplanken, B., & Maio, G. R. (2012). Communal narcissism. *Journal of Personality and Social Psychology, 103*(5), 854–878.
<https://doi.org/10.1037/a0029629>
- He, J., Van de Vijver, F. J. R., Fetvadjev, V. H., de Carmen Dominguez Espinosa, A., Adams, B., Alonso-Arbiol, I., Aydinli-Karakulak, A., Buzea, C., Dimitrova, R., Fortin, A., Hapunda, G., Ma, S., Sargautyte, R., Sim, S., Schachner, M. K., Suryani, A., Zeinoun, P., & Zhang, R. (2017). On enhancing the cross-cultural comparability of Likert-scale personality and value measures: A comparison of common procedures. *European Journal of Personality, 31*(6), 642–657. <https://doi.org/10.1002/per.2132>

- Roccas, S., & Brewer, M. B. (2002). Social identity complexity. *Personality and Social Psychology Review*, 6(2), 88–106. https://doi.org/10.1207/S15327957PSPR0602_01
- Roccas, S., & Schwartz, S. H. (1993). Effects of intergroup similarity on intergroup relations. *European Journal of Social Psychology*, 23(6), 581–595.
<https://doi.org/10.1002/ejsp.2420230604>
- Snyder, M., & Cantor, N. (1998). Understanding personality and social behavior: A functionalist strategy. In D. T. Gilbert, S. T. Fiske, & G. Lindzey (Eds.), *The handbook of social psychology* (4th ed., pp. 635–679). McGraw-Hill.